# mRNA Vaccines Against COVID-19 as Trailblazers for Other Human Infectious Diseases

**DOI:** 10.3390/vaccines12121418

**Published:** 2024-12-16

**Authors:** Rossella Brandi, Alessia Paganelli, Raffaele D’Amelio, Paolo Giuliani, Florigio Lista, Simonetta Salemi, Roberto Paganelli

**Affiliations:** 1Istituto di Science Biomediche della Difesa, Stato Maggiore Della Difesa, 00184 Rome, Italy; rossella.brandi@persociv.difesa.it (R.B.); direttore@isbd.difesa.it (F.L.); 2Istituto Dermopatico dell’Immacolata, 00167 Rome, Italy; alessia.paganelli@gmail.com; 3Independent Researcher, 00162 Rome, Italy; raffaele.damelio@gmail.com; 4Poliambulatorio Montezemolo, Ente Sanitario Militare del Ministero Della Difesa Presso la Corte dei Conti, 00195 Rome, Italy; paolo.giuliani@esercito.difesa.it; 5Division of Internal Medicine, Azienda Ospedaliero-Universitaria S. Andrea, 00189 Rome, Italy; 6Internal Medicine, Faculty of Medicine and Surgery, Unicamillus, International School of Medicine, 00131 Rome, Italy

**Keywords:** mRNA, vaccines, infectious diseases, COVID-19

## Abstract

mRNA vaccines represent a milestone in the history of vaccinology, because they are safe, very effective, quick and cost-effective to produce, easy to adapt should the antigen vary, and able to induce humoral and cellular immunity. Methods: To date, only two COVID-19 mRNA and one RSV vaccines have been approved. However, several mRNA vaccines are currently under development for the prevention of human viral (influenza, human immunodeficiency virus [HIV], Epstein–Barr virus, cytomegalovirus, Zika, respiratory syncytial virus, metapneumovirus/parainfluenza 3, Chikungunya, Nipah, rabies, varicella zoster virus, and herpes simplex virus 1 and 2), bacterial (tuberculosis), and parasitic (malaria) diseases. Results: RNA viruses, such as severe acute respiratory syndrome coronavirus (SARS-CoV)-2, HIV, and influenza, are characterized by high variability, thus creating the need to rapidly adapt the vaccines to the circulating viral strain, a task that mRNA vaccines can easily accomplish; however, the speed of variability may be higher than the time needed for a vaccine to be adapted. mRNA vaccines, using lipid nanoparticles as the delivery system, may act as adjuvants, thus powerfully stimulating innate as well as adaptive immunity, both humoral, which is rapidly waning, and cell-mediated, which is highly persistent. Safety profiles were satisfactory, considering that only a slight increase in prognostically favorable anaphylactic reactions in young females and myopericarditis in young males has been observed. Conclusions: The COVID-19 pandemic determined a shift in the use of RNA: after having been used in medicine as micro-RNAs and tumor vaccines, the new era of anti-infectious mRNA vaccines has begun, which is currently in great development, to either improve already available, but unsatisfactory, vaccines or develop protective vaccines against infectious agents for which no preventative tools have been realized yet.

## 1. Introduction

Anti-coronavirus disease (COVID)-19 vaccines composed of messenger ribonucleic acid (mRNA) represent the turning point for timely confronting and controlling the COVID-19 pandemic. They were successfully tested and made ready for use with unprecedented speed, in less than one year, thus overcoming difficulties that seemed insurmountable, including not only the essential issues of safety and efficacy, but even the crucial problem of large-scale fast production, in order to quickly immunize hundred millions of people worldwide.

## 2. mRNA

mRNA was discovered in 1961, as the intermediate able to carry the genetic information from deoxyribonucleic acid (DNA) to ribosomes for protein synthesis [1]. This molecule is highly unstable and its function within the cell involves migrating from the nucleus to the cytoplasm. Once in the cytoplasm, it undergoes translation at the ribosomal level to produce proteins, which are then released in/outside the cell or expressed at the cell surface. In 1984, 23 years after mRNA discovery, a group of biologists at Harvard University demonstrated that the mRNA molecule could be synthetized in the laboratory by taking advantage of the use of a viral RNA-synthetizing enzyme [2]. At the same time, Krieg and Melton provided evidence that synthetic RNA functioned in an identical manner as natural, endogenous mRNA by synthesizing the protein whose genetic information it was carrying after being injected in the cytoplasm of frog oocytes [3]. In 1990, the same result was obtained after directly injecting synthetic naked mRNA in the animal muscle [4], and in 1992, it was demonstrated that the direct injection of synthetic naked mRNA in the hypothalamus of rats with diabetes insipidus for the vasopressin defect could reverse the defect by inducing vasopressin synthesis [5]. The demonstration that mRNA could be obtained in the laboratory by in vitro transcription (IVT) and that it functioned exactly like natural mRNA paved the way to its possible uses, including medical purposes. In spite of this progress, the medical application of this discovery has been hindered by several factors. These include the high instability of mRNA [6], its reactogenicity, increased by the presence of double-stranded (ds)RNA, which may contaminate IVT mRNA; its ability to stimulate Toll-like receptor (TLR) 3 [7]; its ability to strongly activate certain TLRs, such as TLR 7, and 8 [8,9]; and its inefficient capacity to enter cells as naked mRNA [10,11]. Therefore, the development of an effective mRNA delivery method is crucial. Overcoming all these obstacles was long and complex; however, breakthrough progress was represented by the observation, in 2005, that mammalian mRNA molecules containing modified nucleosides, such as pseudo-uridine and some methylated nucleosides, are poorly recognized by the innate immune system [12]. Moreover, a careful purification of the IVT mRNA by selective dsRNA binding to cellulose in an ethanol-containing buffer may eliminate nearly all dsRNA, thus markedly reducing the activation of the innate immune system [13]. Highly purified, nucleoside-modified mRNA is a stable and poorly reactogenic molecule that may also markedly increase the quantity of protein production up to 1000-fold in primary human dendritic cells [11]. Such discoveries, which have made the medical use of synthetic mRNA possible, were considered so crucial that the Nobel Prize for Physiology or Medicine 2023 was awarded to Katalin Karikò and Drew Weissman “for their discoveries concerning base modifications that enabled the development of effective mRNA vaccines against COVID-19”.

Synthetic mRNA is composed of basic structural elements of mature eukaryotic mRNA, including the five-prime cap (5′ cap), five-prime untranslated region (5′ UTR), open reading frame (ORF) region, three-prime untranslated region (3′ UTR), and poly (A) tail structure [14] (Figure 1).

A 7-methylguanosine (m7G) cap is present at the 5′ end of the mRNA sequence, which enhances the mRNA’s stability, increases protein translation via binding to eukaryotic translation initiation factor 4E (EIF4E), and makes mRNA not recognizable by the host innate immune system [15,16]. UTRs play a role in the efficiency of replication and translation of mRNA by interacting with RNA-binding proteins [17]. The introduction of UTRs from highly expressed genes, such as α- and ß-globin, at 5′ and 3′ UTRs allows us to stabilize mRNA and regulate subcellular localization [17]. The length of the poly(A) tail enhances stability and increases the half-life of mRNA by indirectly controlling mRNA translation and the RNA exonuclease degradation process [16,17]. The yield of the pivotal region of the ORF in the mRNA vaccine, responsible for carrying the genetic information encoding the target antigen, can be enhanced by replacing infrequently used codons with more commonly occurring ones that encode the same amino acids. Even though such modifications should be carefully implemented, considering that they may have a negative impact on proper protein folding [17]. All these studies and discoveries paved the way to the rapid development of mRNA vaccines [6,18]. Their unusually swift development during the emergence of the COVID-19 pandemic marked the first instance in the history of pandemics where safe and effective vaccines could be promptly provided [19]. Traditional vaccines, composed of living or killed whole microorganisms or purified or recombinant subunits [20], are poorly amenable to be rapidly adapted to circulating microorganisms in case of the sudden burst of a pandemic, like COVID-19 or influenza. This has hindered the timely development of an effective vaccine in influenza pandemics [19], whereas with COVID-19, innovative vaccines, such as mRNA vaccines, made the difference because they have an easily adaptable platform. Compared to traditional vaccines, mRNA vaccines provide a number of clear benefits. Firstly, they have the potential to efficiently and stably encode and express a wide range of proteins, and their sequence can be easily modified to optimize vaccine development, thus saving time and costs [11]. Additionally, the production and purification processes for mRNA vaccines are relatively safe (mRNA is not infectious) and uniform, allowing for the potential standardization and streamlined development of similar vaccines [11,16]. Moreover, mRNA vaccines can express target proteins more effectively than DNA-based vaccines and are generally considered to be at lower risk of integration into the host DNA, given their short permanence in the host and the unlikely entering into the cell nucleus [21]. In addition, mRNA vaccines induce the synthesis of an antigenic protein in the host cells that is identical to the one derived from the infectious agent [22]. Lastly, these vaccines stimulate innate immunity and robust neutralizing antibodies (nAbs) as well as specific cellular immunity [19]. This happens without the need of an adjuvant, considering that mRNA-LNP vaccines are immune-stimulating per se. By inducing the synthesis of an identical protein antigen by human cells, mRNA vaccines may be assimilated in vaccinology to what monoclonal antibodies have represented in serology and passive immunotherapy. In fact, both are biologically obtained, thus exactly mimicking the natural process; thus, they noticeable reduce the time and the cost for production since they avoid all purification steps, with their burden of possible mistakes or contaminations, because they are no longer necessary. Table 1 reports a comparison of several characteristics between traditional and mRNA vaccines.

Self-amplifying mRNA vaccines have also been developed by inserting replicase genes from alphaviruses encoding RNA-dependent RNA polymerase. These vaccines may strongly stimulate the innate and adaptive immune system using very low mRNA doses, because they may replicate intracellularly [17]. A series of experimental studies indicate this as a promising avenue in humans too, provided that safety issues are checked [11].

Recently, circular (non-linear) mRNA vaccines have been developed. They are more stable and have shown to be able to induce more durable synthesis of a larger quantity of antigen, represented by the trimer receptor binding domain (RBD) of the Spike (S) protein of the severe acute respiratory syndrome coronavirus (SARS-CoV)-2, than the linear nucleoside-modified ones, thus eliciting broad and robust nAbs and T-cell immunity in mice and monkeys [23]. Due to its covalently closed structure, circular RNA is protected from exonuclease degradation [24]; moreover, circular RNA has a higher thermostability than the nucleoside-modified mRNA, being able to maintain cell transfection efficiency even after two weeks at room temperature [23]. Lastly, unmodified circular RNA is unable to activate innate immunity through TLRs and retinoic acid-inducible gene (RIG)-I pathways both at the cellular level and in mice, unlike unmodified linear RNA [25]. This is a very important point, considering that nucleoside modification was the only obligate strategy to make linear mRNA almost invisible to the host immune system; however, such a winning strategy may have some undesired consequences, as the induction of T-regulatory cells [26], which may reduce vaccine reactogenicity, probably also reduces the specific antibody response. Although highly promising, circular mRNA vaccines are still in the pre-clinical phase of study.

## 3. Delivery Systems

Regarding the delivery system, early studies in the sixties and seventies of the last century tested the efficiency of liposomes in delivering therapeutics and vaccines and, in 1978, even mRNA. However, a change in pace began in 2001, with the first testing of the four-component lipid nanoparticles (LNPs) to deliver DNA. In Figure 2 the main steps to the development of mRNA vaccines are reported.

The four components are represented by a cationic lipid component, which allows mRNA release in the cytoplasm; a cholesterol component, which is a structural component that contributes in stabilizing the LNP; the poly-ethylene-glycol (PEG)-linked lipid component, which increases the half-life of the nanoparticles; and the helper phospholipids, which are relevant for the bilayer structure of the particle and may influence target organ specificity [11,14,27].

However, the cationic lipid component, although highly effective, is pro-apoptotic and pro-inflammatory as well; thus, it has been replaced by the ionizable lipid component, which is better tolerated, while maintaining high effectiveness [17]. The ionizable lipid component has the characteristic of being uncharged, thus making it neutral at a pH higher than the acid dissociation constant (p*K*_a_) and thus making it non-toxic and facilitating its systemic circulation. Conversely, in conditions of pH lower than the p*K*_a_, it becomes cationic and is able to create a gradient with the anionic mRNA. Such behavior allows a better complexation and encapsulation of mRNA and facilitates, after the endocytosis, the endosomal membrane fusion and the consequent cytoplasmic release. Examples of ionizable components with suitable p*K*_a_ ranging from 6 to 7 are those used in the approved COVID-19 mRNA vaccines [28].

Cholesterol (but even oxidized cholesterol derivatives and phytosterols) is a structural lipid that contributes to the bilayer stability and increases its rigidity, thus preventing the leakage of the transported therapeutic/vaccine [28,29]. It has been observed that the incorporation of C-24 alkyl phytosterols into LNPs (eLNPs) enhances gene transfection [30].

The PEG-linked lipid component stabilizes the colloid and prolongs the half-life of the circulating nanoparticles. Moreover, PEG lipids prevent the physical aggregation of the LNPs, thus increasing their storage stability [28].

The helper phospholipids, as 1,2-distearoyl-sn-glycero-3-phosphocholine (DSPC) present in the LNPs of the two COVID-19-approved mRNA vaccines, provide bilayer structural stability and play a role in the biodistribution of LNPs.

## 4. SARS-CoV-2

SARS-CoV-2 is a single-stranded, positive-sense RNA virus pertaining to the family of *Coronaviridae*, subfamily *Orthocoronavirinae*, which is formed by four genera: the Alpha, Beta, Gamma, and Delta coronaviruses, with the first two only being able to infect and be pathogenic for humans [31]. Since the virus is equipped with a low-fidelity RNA polymerase prone to transcription mistakes, SARS-CoV-2 continuously mutates, causing the emergence of many variants around the world. In the first year of its worldwide spreading, many variants of concern (VOCs) and variants of interest (VOIs), according to the WHO classification [32], were described. Five VOCs have been identified, the first four of which, Alpha, Beta, Gamma, and Delta, were reported in 2020 and subsequently disappeared, to be replaced in 2021 by the VOC Omicron with its several sublineages (from BA.1 to BA.5). New Omicron recombinant lineages, deriving from BA.2, were first identified in Southeast Asia and rapidly spread worldwide [32,33,34]. Several VOIs have also been described and have been named after the Greek alphabet as variants Epsilon, Zeta, Eta, Theta, Iota, Kappa, Lambda, and Mu [32]. On 15 March 2023, the WHO’s variant-tracking system considered the classification of Omicron sublineages independently as variants under monitoring (VUMs), VOIs, or VOCs. Moreover, the WHO announced that it would maintain the Greek alphabet for VOCs only but not for VOIs, which should only be denominated according to the classification tools Nextstrain and Pango. These tools classify SARS-CoV-2 sequences according to the hierarchical structure—like a family tree approach and genetic relatedness. Since the VOIs also disappeared, all of them were designated “previous VOIs” in 2021, except Lambda and Mu, which were designated “previous VOIs” on 9 March 2022 [32]. Currently, the WHO is actively monitoring various SARS-CoV-2 variants and has de-escalated BA.2, BA.4, and BA.5 from its list of SARS-CoV-2 VOCs after observing a prevalence of less than 1% for more than eight epidemiological weeks globally and across the WHO regions: as of 28 June 2024, circulating VOIs are BA.2.86 and JN.1, whereas as of 24 September 2024, VUMs are JN.1.7, KP.2, KP.3, KP.3.1.1, JN.1.18, LB.1, and XEC [35]. JN.1 emerged in late 2023 and is characterized by the S protein mutation Leu455Ser, which is associated with an increased immune evasion [36]. The emergence of the variants is a consequence of the high mutations rate of the genome; a mutation rate of 1.1 × 10^−3^ mutations/site/year has been calculated for SARS-CoV-2 between 2019 and 2020 [37,38], and then the mutation rate gradually decreased (6.677 × 10^−4^ mutations/site/year) during the subsequent evolution of the infection process [39]. The mutations are generally due to the pressure of a moderate, inefficient immune response, inability to clear the virus, and focus on one or two protein antigens, as it has been observed in immunocompromised patients [40,41]. However, variants appear not only as a consequence of mutations but also of recombination [42]. The rate of SARS-CoV-2 antigenic drift is influenced by various factors, including the viral polymerase mutation rate, the capacity of the viral protein to accept amino acid substitutions that reduce the receptor binding while maintaining the viral fitness, the number of virions involved in and generated during the infection, and the immune selection pressure on the antigenic sites. This drift occurs at the S protein level with surprising rapidity and frequency approximately 10-fold higher, in the case of the Omicron variant, than for influenza A HA [43]. Such high variability and the surprisingly high mutation rate cast serious doubts on the maintenance of efficacy of monoclonal antibodies and vaccines, considering that some mutations are able to modify the antigenic viral structure, thus activating immune escape mechanisms [44].

In the Alpha VOC (B.1.1.7, according to the Pango lineage, also known as GRY [formerly GR501Y.V1], according to the Global Initiative on Sharing All Influenza Data [GISAID] clade, and 20I/501Y.V1, according to the Nextstrain clade), identified in the UK in September 2020 and designated as a VOC on 18 December 2020 [45], eight mutations in the S protein have been observed, one of which, mutation N501Y, in the RBD [37], is able to increase transmissibility [46,47] (Table 2). The B.1.1.7 variant spread globally and, in the strains isolated in the UK and USA, mutation E484K was acquired, associated with an estimated 6-fold decrease in viral sensitivity to the sera of subjects immunized with the Pfizer BNT162b2 mRNA vaccine and 11-fold to the sera of convalescent individuals [44,47].

In the Beta (B.1.351, according to the Pango lineage, also known as GH/501.V2, according to the GISAID clade, and 20H/501Y.V2, according to the Nextstrain clade) and Gamma (P.1, according to the Pango lineage, also known as GR/501Y.V3, according to the GISAID clade, and 20J/501Y.V3, according to the Nextstrain clade) VOCs, first documented in South Africa and Brazil, in May and November 2020, and designated as VOCs on 18 December 2020 and 11 January 2021, respectively [43], characterized by 8 and 12 mutations in the S protein [48], respectively, the additional mutations E484K and K417 at the RBD site were acquired. The variants carrying all the three mutations, N501Y, E484K, and K417N/T together, showed a significantly reduced susceptibility to vaccinated and convalescent sera [49].

The Delta (B.1.617.2, according to the Pango lineage, also known as G/452R.V3, according to the GISAID clade, and 21A/S:478K, according to the Nextstrain clade) variant, first isolated in India in October 2020 and designated as a VOC on 11 May 2021 [45], is characterized by nine mutations in the S protein [48]. It shares mutation D641G with the Alpha, Beta, and Gamma VOCs, and in some subvariants, mutation K417N. Additionally, mutations T478K, P681R, and L452R appeared de novo in this variant. The increase in transmissibility for the Alpha, Beta, Gamma, and Delta VOCs has been calculated in 29%, 25%, 38%, and 97%, respectively, thus providing the explanation for their quick diffusion [50]. Immune evasion, as documented by the reduced sensitivity to nAbs in sera obtained from convalescent patients from infection with ancestral strain, was 5.7-fold, 8.2-fold, and 2.3-fold in the Delta, Beta, and Alpha variants, respectively, compared with the Wuhan ancestral strain [51].

The Omicron (B.1.1.529, according to the Pango lineage; GR/484A, according to the GISEAD clade; and 21K, according to the Nextstrain clade) variant, first described in various countries in November 2021 and designated as a VOC on 26 November 2021, has over 30 mutations at the S protein level. It is reported to be more transmissible but able to induce less severe disease than the Delta variant [52], and it is poorly recognized by neutralizing antibodies elicited by mRNA vaccines. In fact, nAbs elicited by Moderna mRNA-1273 and Pfizer-BNT162b2 vaccines, compared to the ancestral Wuhan strain, show a 39- and 37-fold reduction in activity, respectively [53]. These data and the marked immune evasion shown with therapeutic monoclonal antibodies allow us to conclude that the Omicron variant has undergone a real antigenic shift, not dissimilar from the ones undergone by influenza virus, which were responsible for pandemics [53]. The immune evasion was lower if vaccinated subjects had previously had SARS-CoV-2 infection [53], thus underlying the reciprocal synergic protection induced by natural infection and vaccination. It has been observed that the Omicron variant may infect and grow in the human bronchus 70-fold faster than the Delta variant and the ancestral strain [54]. Sera from subjects immunized with two doses of Pfizer-BNT162b2 could neutralize the Omicron variant 22-fold less than the ancestral Wuhan viral strain, and only a third vaccine dose could restore a neutralizing activity similar to the one against the Wuhan strain [55] (Table 2).

Moreover, sera from subjects immunized with two doses of Pfizer-BNT162b2 or Moderna mRNA-1273 had absent or very low levels of neutralizing activity against the Omicron variant compared with the Wuhan strain, whereas neutralizing activity of sera from individuals who had been exposed to the S protein three or four times through both infection and vaccination was maintained, although at significantly reduced levels [56]. Currently, bivalent vaccines are used: in the composition, both the ancestral and the circulating mutated strains are used [57]. However, the preliminary studies on bivalent vaccines containing the ancestral viral strain together with the Omicron subvariant BA.1 showed only a modest increase in nAbs against BA.1 itself, of the order of 1.5–1.75-fold, probably for the “original antigenic sin” effect, that is, the imprinting with the ancestral Wuhan viral strain conditioning any future immune response in a dominant way [19,58]. This situation increased the lack of confidence in the effectiveness of COVID-19 vaccines and the hesitancy in the general population, already triggered by concerns of safety, due to the very quick approval, which could have prevented the observation of possibly late-occurring adverse events, so that by mid-November 2022 in the USA, only 10% of people were vaccinated with the bivalent ancestral and BA.4/BA.5 vaccines. However, by December 2022, BA.4 had disappeared from the USA, and BA.5 was identified in less than 25% of the circulating strains and was replaced by the new subvariants, such as BQ.1, BQ.1.1, BF.7, XBB, and XBB.1 [58]. Nevertheless, bivalent vaccines containing BA.4/BA.5 Omicron subvariants demonstrated similar effectiveness against infections, hospitalizations, and death even in the period when BA.4 and BA.5 were largely replaced by BQ.1 and BQ.1.1 and later on by XBB and XBB.1.5 [59]. Based on the high viral variability and the higher and broader protection afforded by the bivalent vaccines, the approach of the regulatory agencies, such as the FDA and EMA, is to periodically monitor the circulating viral strains, in order to give recommendations on the updated vaccine composition, with a model similar to the one adopted for influenza vaccine [60]. For the reasons reported above and for the short-term effectiveness of COVID-19 mRNA vaccines against asymptomatic and/or mild infections [61], the virus is still widely circulating; therefore, a transition from the pandemic to the endemic state has occurred, while the efficacy of vaccines was almost exclusively expressed in the prevention of severe disease, hospital and ICU overcrowding, and deaths [19,61]. Moreover, more and more frequently, fully vaccinated individuals may become a source of transmission; paradoxically, in a nosocomial outbreak in Israel originated by a fully vaccinated subject, fourteen severe cases or deaths have been described in fully vaccinated individuals, whereas two unvaccinated subjects had a mild infection [62].

The ease of mutation shown by the SARS-CoV-2, particularly with the Omicron variant and subvariants, questions the possibility of adapting the vaccine to the circulating viral strain, thus forcing the vaccine to chase the virus in a condition of partial mismatch instead of anticipating it in the best conditions of full vaccine/virus matching. The high viral variability and the poor vaccine effectiveness against asymptomatic and/or mild breakthrough infections are the elements that do not allow the virus to be eradicated; rather, it appears that the pressure of the vaccine-stimulated immune response forces continuous modifications of the virus antigenic structure, thus inducing antigenic drifts [43] and perpetuating its circulation in a form that is always poorly recognizable by the immune system. This has been confirmed very recently, by observing the reduced effectiveness against infections, hospitalization, or death associated with the updated vaccine containing the Omicron subvariant XBB.1.5, when the circulating viral subvariant XBB.1.5 has been replaced with the subvariant JN.1, thus indicating the need to further update the vaccine with JN.1 replacing XBB.1.5 [63]. However, already in May 2023, a WHO statement indicated the need for a monovalent SARS-CoV-2 vaccine containing the variant XBB.1.5 [64], more recently replaced with the currently prevalent variant JN.1 [65].

## 5. mRNA Vaccines Against COVID-19

### COVID-19

The first mRNA vaccines approved are those against COVID-19, which represent a milestone in the history of vaccinology. Two mRNA COVID-19 vaccines have been authorized, namely the Pfizer-BNT162b2 and the Moderna mRNA-1273 (trade names: Comirnaty and Spikevax, respectively), which have been successfully tested in hundred millions of people worldwide from the end of 2020 onward [66]. Both vaccines are very similar, considering that they contain mRNA encoding the trimeric S protein of the virus, which forms the virus corona and allows the virus to enter the host cells by interacting, at the level of the RBD, with the angiotensin-converting enzyme 2 (ACE2) receptor and taking advantage of the transmembrane protease, serine 2 (TMPRSS2), an enzyme colocalized with ACE2 at the cell membrane, which works as the dominant proteolytic driver of S protein activation [67] (Figure 3).

Both vaccines use nucleoside-modified mRNA, with uridine being replaced by N1-methylpseudouridine, encoding the full-length S protein (both subunits, the RBD-containing S1 and S2, involved in the fusion with the cell membrane) of the SARS-CoV-2 with 2 proline substitutions (S-2P, K986P, and V987P), so that the encoded protein is stabilized in its prefusion conformation [68]. Conversely, CVnCoV, an LNP-mRNA COVID-19 candidate vaccine developed by CureVac, does not contain modified nucleosides but sequence-engineered, guanine-cytosine-enriched mRNA. The CVnCoV vaccine has achieved positive preclinical results by inducing a robust humoral and cell-mediated immune response [69]. The Phase 1 dose-escalation clinical trial was also successful by showing the safety and immunogenicity of CVnCoV administered to 245 volunteers, to whom two doses of 2, 4, 6, 8, and 12 μg of vaccine were administered 4 weeks apart [70]. Despite these promising premises, Phase 2b/3 of the randomized, observer-blinded, placebo-controlled HERALD clinical trial of two doses, 4 weeks apart, of 12 μg of the CVnCoV candidate vaccine showed an overall vaccine efficacy against symptomatic COVID-19 of only 48.2%. This unsatisfactory result induced CureVac “to cease activities on the CVnCoV candidate and to focus efforts on the promising and rapidly progressing development of the next generation vaccine candidates” [71], one of which, the CV2CoV candidate, had already shown to be able to induce strong humoral and cellular immune responses in non-human primates [72].The mRNA amount in the Pfizer-BNT162b2 was 30 μg/dose with two doses intramuscularly administered 3 weeks apart [73], whereas in the Moderna mRNA-1273, the mRNA quantity was 100 μg/dose, and two doses were intramuscularly administered 4 weeks apart [74]. In both vaccines, dsRNA was removed in order to reduce reactogenicity, and LNPs are used as the delivery system, based on the four components, including the ionizable lipid component, which may further increase vaccine performance. These vaccines are not adjuvanted, considering that both mRNA and LNPs act as adjuvants. The Comirnaty has, as a cationic ionizable lipid, the ((4-Hydroxybutyl)azanediyl)bis(hexane-6,1-diyl)bis(2-hexyldecanoate), ALC-0315, with a pK_a_ of 6.09, whereas the Spikevax has (Heptadecan-9-yl 8-((2-hydroxyethyl) (6-oxo-6-(undecyloxy) hexyl) amino) octanoate}, SM102, with a pKa of 6.75 [28]. Both have DSPC as helper phospholipids and cholesterol as a sterol component. Pfizer-BNT162b2 requires −80 °C for storage with a shelf life of up to 6 months, whereas Moderna mRNA-1273 may be stored at −20 °C for the same period [75].

The efficacy of Pfizer-BNT162b2, explored in a pre-registration Phase 1/2/3 study on more than 43,000 healthy subjects, was 95% 7 days or more after the second vaccine dose was administered to subjects without prior evidence of infection and 94.6% in both those without and those with prior evidence of infection [76]. The most frequent local reaction was pain, more often after the first dose than the second, observed in a percentage of 70–80%, without a significant difference between the 16–55-year-old and the >55-year-old groups. Systemic reactions were represented by fatigue, headache, chills, and muscle pain, more frequently observed following the second vaccine dose. However, most reactions were mild to moderate [76]. The follow-up was 2 months in half of the cases and 3.5 months in a smaller number of cases, instead of the originally planned 2 years, because, in light of good safety and excellent efficacy, it was considered unethical to continue administering placebo for 2 years to a considerable number of subjects, preventing their possibility to be vaccinated in the attempt to arrest the spreading of the pandemic, particularly the more severe clinical forms, those causing the congestion of hospitals and intensive care units (ICUs), with the consequent collapse of national health systems and death. Even the model of a continuous Phase 1/2/3 study allowed us to gain time and contributed to realizing the near-miracle of a ready-to-use vaccine in less than one year.

The safety and efficacy of the Moderna mRNA-1273 vaccine, investigated in a Phase 3 study on a population of over 30,000 subjects, are very similar to those reported above with the Pfizer-BNT162b2 vaccine. The most frequent local reaction was local pain, whereas the most frequent systemic reactions were headache, fatigue, fever, chills, and arthro-myalgias. The efficacy was 94.1% [77] (Table 3).

Both the Comirnaty and Spikevax vaccines received Emergency Use Authorization (EUA) from the Food and Drug Administration (FDA) and the European Medicines Agency (EMA) on 11 December 2020–21 December 2020 and 18 December 2020–6 January 2021, respectively, to be administered to 16-year-olds and older and 18-year-olds and older, respectively. Although the pediatric population is not particularly exposed to the risk of COVID-19, severe cases of COVID-19 have been observed, particularly in children older than 10 years [78], and complications, such as the potentially lethal multisystem inflammatory syndrome in children and long COVID [79] may have a significant impact on this population. Thus, based on a series of favorable pre-registration studies that investigated safety, immunogenicity, and efficacy in these pediatric populations [80,81,82,83,84,85], the FDA and EMA approved the mRNA vaccines even for the age range of 12–16 and 12–18 years, and then for the age range of 5–11, and finally even for the age range of 6-month–4-year-olds for Pfizer BioNTech and 6 months–5 years for Moderna. In this last age range, three doses of 3 μg/dose were administered 3 weeks apart for the first two doses, 8 weeks apart for the second and third dose for Pfizer BioNTech, and two doses of 25 μg/dose were administered 4 weeks apart for the Moderna vaccine. In the age range of 5–11 years, the amount of mRNA per dose was 10 μg for Pfizer BioNTech and 50 μg per dose for Moderna, whereas adolescents received a dose amount equal to that for adults. In the age range of 6 months–5 years, the mRNA vaccines were safe, with very few severe adverse events [86], and effective [87], without any significant difference between the two vaccines. Even in the age range of 5–11 years, the mRNA vaccines were safe, with generally local adverse events, whereas severe adverse events were rare. Moreover, mRNA vaccines were effective in preventing SARS-CoV-2 infection and severe COVID-19-related illnesses [88]. Vaccinated adolescents reported health events more frequently following the second mRNA COVID-19 vaccine dose, while younger age groups did not report events more frequently than their unvaccinated counterparts [89].

## 6. Immune Response to SARS-CoV-2 and to COVID-19 Vaccines

Once intramuscularly injected, mRNA vaccines may transfect different types of cells, including somatic cells (such as muscle cells and epidermal cells), immune cells at the injection site, and immune cells in the secondary lymphoid organs, such as draining lymph nodes and spleen. The antigen produced by transfected non-immune cells, such as muscle cells, is subsequently degraded by proteasomes, and it may then be complexed with the major histocompatibility complex (MHC) type I to be presented to CD8^+^ cytotoxic T-cells. Transfected muscle cells may even activate and recruit bone-marrow derived dendritic cells, to which they may transfer, through a not well-known mechanism, the antigen [90]. However, the main cellular pathway for activating the adaptive immune response is through mRNA-LNP vaccine internalization and translation by antigen-presenting cells at the injection site and the draining lymph nodes [91], as shown in Figure 4A. Antigen-presenting cells in the secondary lymphoid organs bridging innate and adaptive immunity trigger molecular events leading T-follicular helper lymphocytes, a type of T helper (Th) cells present in the germinal centers (GCs), to express either the CD4^+^ interferon-γ producing Th-1 or the interleukin (IL)-4-producing the Th-2 cell phenotype. This induces B-cells in the GCs to undergo somatic hypermutation (SHM), increased affinity, and progressive maturation, which leads to the generation of antibody-secreting long-lived plasma cells (PCs) and affinity-matured memory B-cells [91] (Figure 4B).

Th-1 lymphocytes are very important because, like CD8^+^ cytotoxic T-lymphocytes (CTL), they are specialized in fighting intracellular microorganisms, such as viruses, whereas Th-2 cells are relevant for defense against helminths and parasites. Long-lived PCs may only be obtained after convenient B-cell maturation inside the GCs and are needed for a robust nAb response that remains quantitatively stable even in the absence of the antigen and probably for a favorable environment in the niche in the bone marrow, where they are housed [92], or in consequence of B-cell polyclonal activation [93].

The antibody response to exogenous antigens should mature in order to achieve the best match to neutralize the invading organism. Affinity maturation occurs in GCs, which are transient microstructures formed within the follicles of secondary lymphoid tissues in response to certain types of immunization and foreign pathogens. A mature GC comprises two functionally distinct compartments, a dark zone (DZ) and a light zone (LZ), which are microanatomical substructures located in the follicles of secondary lymphoid organs (Figure 4B).

The receptors of B-lymphocytes recognize the foreign antigen enter the GC and proliferate, while SHM induces the affinity maturation of their B-cell receptor. The best matching receptor-carrying B-cells are selected and differentiated into long-lived bone marrow PCs and circulating memory B-cells (Figure 4B). Long-lived PCs can be retained in the tissue of origin, such as the skin, the respiratory tract, or the gut-associated lymphoid tissue. Tissue-resident memory B-cells are readily available effector B-cells mediating long-term protection against a pathogen, and vaccines are meant to induce their development.

Th cells promote the activation of antigen-specific B-cells through contact from the T-cell area in lymphoid follicles and migration to populate the center of the follicle, attracted to the DZ, where stromal cells induce a first round of affinity maturation through SHM and proliferation. Then, a transition to the LZ of the GC occurs, where follicular dendritic cells re-expose the antigen in the presence of T-follicular helper cells, promoting further maturation and selective expansion, and class switch recombination may also take place; thereafter, B-lymphocytes return to the DZ, and further SHM/proliferation occurs, exiting from the GC as antibody-producing PCs or memory B-cells [94,95,96].

The early waning of nAbs in COVID-19 infection might be due, at least in part, to an extrafollicular B-cell activation pathway, which, at variance with the one involving a GC reaction, induces a type of B-cell response associated with less SHM, a lack of generation of long-lived PCs, and robust production of broad, short-term antibodies. These are associated with the simultaneous appearance of autoantibodies, as it has been observed in active Systemic Lupus Erythematosus (SLE) [97], particularly directed against nuclear and carbamylated antigens, although for a short time, in a locally highly inflamed environment [98]. Loss of Bcl-6^+^ T-follicular helper cells and GCs has been found in patients with severe COVID-19 [99]. Conversely, in mRNA-vaccinated subjects, the humoral immune response is principally generated in GCs, which are well represented in lymph nodes [100], where the S protein and the mRNA have been found up to two months post vaccination [101], even though an in vivo study of lymph node kinetics after mRNA vaccination revealed marked heterogeneity in the GC response of vaccinated healthy subjects, with one third showing the absence of GCs after 29 weeks [102]. However, B-cell maturation through the GC pathway in most vaccinated individuals should stimulate a durable antibody response, which, instead, is quickly waning [101], and is associated with inflammation and autoreactivity [103]. Indeed, both vaccine components, LNP and mRNA, may activate the inflammatory pathways, thus behaving as adjuvants: LNP through its ionizable lipid component and the induction of the pro-inflammatory cytokine IL-6, which strongly stimulates humoral immunity [104]; and mRNA through TLRs, particularly TLR-7, and anti-viral pathways, particularly those mediated by the RIG-1-like receptor melanoma differentiation-associated protein 5 (MDAP5), to further polyclonally enhance humoral immunity [105]. An inflammatory environment induced by the adjuvant-like activity of the LNP-mRNA vaccines is, therefore, useful for stimulating a potent nAb response, which is often associated with autoreactivity [103]. mRNA vaccine administration is associated with a robust specific T-cell response, in which T-follicular helper cells appear skewed toward a CD4^+^ Th 1 type, a cell population that, in synergy with specific CD8^+^ CTL, may exert an effective defense against the viruses, which are intracellularly protected from the action of antibodies [91]. In vaccinated normal subjects, S-specific CD4^+^ T-cells at six months post the first dose mostly presented a pattern of central memory and effector memory, whereas S-specific CD8^+^ T-cells also included a significant fraction of terminally differentiated effector cells [106]. This type of response is accompanied by high and stable levels of CD4^+^ T_(SCM)_ stem cell memory, which should assure long-lived cellular protection for decades [106]. The vaccination of subjects who had previously had COVID-19 could induce the most robust antibody response, higher than the one observed in vaccinated subjects not previously infected [97]. Conversely, such infection-mediated, higher immune stimulation could not be observed with a bivalent mRNA vaccine against the Omicron variant for both bnAbs and SARS-CoV-2-specific CD8^+^ T-cells [107]. mRNA vaccines, therefore, represent a real step forward compared with traditional vaccines both for their relatively quick, easy, and cost-effective production, and because they are able to stimulate an excellent adaptive, humoral, and cellular response, together with an innate one, providing protective immunity, particularly following the second vaccine dose [101]. This may not be observed with a traditional recombinant RBD vaccine equipped with an MF-59-like adjuvant [91]. Moreover, the nAb response is broader and mainly formed by IgG in vaccinees in SARS-CoV-2-infected patients, in whom IgM and IgA predominate, particularly against the Wuhan strain, compared with the antibody response induced by vaccination, which is more protective against the Omicron subvariants [97], thus indicating that an mRNA-LNP vaccine provides a qualitatively better immune response than either natural SARS-CoV-2 infection or traditional vaccines.

Antibodies against a specific site on the N-terminal domain of the SARS-CoV-2 S protein were found to directly augment the binding of ACE2 to the S protein, consequently increasing SARS-CoV-2 infectivity in patients with SARS-CoV-2 infection [108]. High levels of enhanced antibodies were observed in severe COVID-19 and lower levels in patients with milder COVID-19, thus suggesting that even in SARS-CoV-2 infection, an antibody-dependent enhancement (ADE) phenomenon may be present, even though, unlike the ADE described for dengue [109], SARS-CoV [110] and Middle East Respiratory Syndrome (MERS)-CoV [111] are Fc-independent. No antibody-dependent enhancement (ADE) has been observed after vaccination; instead, such a phenomenon has been reported in other viral vaccinations, such as dengue, RSV, and measles inactivated vaccines; feline, simian, and human immunodeficiency virus vaccines; and SARS-CoV and MERS-CoV vaccines [19].

The cause of the rapid waning of nAbs in fully vaccinated people is currently not well understood, considering that B-cell activation and differentiation principally occur through the GC pathway, thus theoretically being able to induce durable bnAbs, provided that the GCs are durable as well [94]. Nonetheless, despite that vaccinated individuals should be able to synthetize antibodies through the GC B-cell maturation pathway to produce long-lived PCs, for unknown reasons, no antibody-secreting cells for SARS-CoV-2 among long-lived PCs have been found in the bone marrow of 19 individuals recently immunized with a SARS-CoV-2 mRNA vaccine, whereas they have been found for influenza and tetanus [112]. This result is consistent with what had previously been observed in COVID-19-infected patients [113]. However, a recent study on a small sub-Saharan cohort, in which post-COVID-19 vaccination IgG and IgA antibodies were maintained at high levels for at least one year, suggests that genetic and/or environmental factors may play a role [114]. In addition, a longitudinal study on the antibody response to COVID-19 mRNA vaccines and breakthrough infections during three years observed that, after an initial waning, the antibody response is stabilized and long-lasting [115]. Furthermore, early antibody waning is balanced by the maintenance of an effective, lasting, cell-mediated immunity, which may protect even in the absence of nAbs, as shown, at an experimental level [116], in primary antibody deficiencies [117,118], and in anti-CD20-treated patients, in whom COVID-19 infections without serious complications have been observed [119]. Moreover, seronegative exposed family members and convalescent subjects with asymptomatic or mild COVID-19 exhibited robust T-cell immunity [120]. However, it should be underlined that the kinetics of specific cellular and humoral protective immunity have recently been described as parallel in naïve normal subjects three months after the first vaccine dose of Pfizer-BNT162b2. Only the group characterized by enhanced antibody-neutralizing activity (high responders) also presented increased frequency of central memory T-cells and durable S-specific CD8^+^ T-cell responses, and only this group was protected against breakthrough infections, thus suggesting that in this condition, cell-mediated immunity may not replace the lacking antibodies in protecting against breakthrough infections [121].

The monitoring of humoral and cellular T- and B-cell memory up to 8 months post-COVID-19 infection showed that immune memory was measurable in at least three out of the four explored immunological compartments, represented by antigen-specific antibodies, B-cells, CD4^+^ T-cells, and CD8^+^ T-cells, 5–8 months post symptom onset in approximately 95% of subjects, thus indicating that durable immunity is a possibility for a large majority of people. Nevertheless, the magnitude of adaptive immune responses to SARS-CoV-2 was quite heterogeneous, thus underlining the still incomplete knowledge of the factors able to influence the protective immunity against SARS-CoV-2 [122].

First-level protection from the infection is afforded by mucosal IgA antibodies; although the mRNA vaccine is not able to directly induce detectable salivary IgA antibodies, they may be produced by memory B-cells attracted to the inflammation site [123]. The limited induction of polyfunctional tissue-resident memory T-cells [124] and of secretory IgA [125] against SARS-CoV-2 in the respiratory mucosa by mRNA vaccination may be in relation to the appearance of breakthrough infections in vaccinated subjects [126,127].

## 7. Factors Influencing the Durability of Vaccine-Induced Protection: Immune Correlates of Protection

Although the mRNA platform is fully innovative and has been shown to be highly effective and safe, the success of a vaccine depends not only on the platform but also on a series of other vaccine-related and vaccinee-related factors [128] (Table 4).

Vaccine-related factors are the vaccine platform, the chemical structure, the antigen dose, and the vaccine schedule. According to the vaccine platform, live vaccines, virus-like particles (VLPs), and the innovative mRNA vaccines are the most effective, as they are able to stimulate a high-quality B-cell response, are long-lasting in live-attenuated vaccines and VLPs, and exhibit early waning in mRNA vaccines [128]. According to the chemical structure, polysaccharide antigens may directly activate B-cell compartments without any T-cell help, thus generating less effective and short-lasting, predominantly oligoclonal, antibodies [129], whereas they may only acquire the capacity to stimulate a T-cell compartment if conjugated to a protein matrix [20,128]. The particulate multimerized high-valency antigen is highly stimulating, able to easily recruit cognate B-cells to drive them toward extrafollicular and GC maturation pathways [130], as well as toward prefusion conformation, which makes accessible neutralization-sensitive epitopes, such as in RSV [131] and SARS-CoV-2; similarly, the antibodies recognizing the trimer HIV envelope in prefusion-closed conformation are broadly neutralizing [132]. These pre- and post-fusion conformations as well as the open or closed conformation dictate whether the epitopes able to stimulate a protective nAb response are made available to be recognized by the immune system; these characteristics, together with the high variability of these RNA viruses, have been at the root of the difficulty to find safe and effective vaccines against them. Thus, RSV, SARS-CoV-2, HIV, and influenza share the high structural variability as a challenge for developing highly effective and protective vaccines. In addition, RSV and SARS-CoV-2 are associated with a prefusion conformation able to expose epitopes recruiting nAbs. In HIV, the envelope trimer in the prefusion conformation is closed instead; thus, the nAb-recruiting epitopes are not exposed. Nonetheless, the antibodies recognizing the tier 2 prefusion-closed conformation are neutralizing and protective [133]. Moreover, immunogenicity may be increased by adjuvants [134], which are used in the less immunogenic, inactivated, and subunit vaccines. Recently, our knowledge on adjuvants was revolutionized by the advent and the application of nanomaterials, so that even the oldest and most used adjuvant, alum, which has traditionally been considered a Th-2-stimulating adjuvant only, when modified as nano-alum, surprisingly becomes able to stimulate an effectively protective Th-1 immune response [135]. The antigen dose is crucial for tuning the recruitment of highly effective, long-lasting T- and B-cells [136]. However, higher doses may also qualitatively modify the humoral immune response. It was observed that after three doses of the COVID-19 mRNA vaccine, there is a marked increase in IgG4 (from 0.04% shortly after the second dose to 19.27% late after the third one). Considering that IgG4 is poorly efficient in complement activation and antibody-mediated cellular phagocytosis, which are critical functions of the antiviral immunity, this may mean a reduced defense associated with a higher antigen dose [137]. Lastly, the route of administration, the dose schedule, and the homologous or heterologous prime-boost plan are pivotal characteristics too. Intradermal and intramuscular routes are more immunogenic than the subcutaneous one, with intradermal showing higher antigen retention in the lymph nodes as well as better antibody and lower CD4^+^ and CD8^+^ T-cell responses than intramuscular administration [138,139]. The type of vaccine schedule for priming may stimulate the growth of the GCs, as recently demonstrated in mice with an HIV env trimer/saponin adjuvant vaccine, whose priming was split into two doses, seven days apart, with the first one consisting of 20% and the second consisting of 80% of the treatment. This two-dose schedule for priming could achieve a 10-fold stimulation of antigen-specific GCs and of serum antibodies as well compared with a bolus administration [140]. Lastly, the heterologous prime-boost seems to be more immunogenic than the homologous one [141].

The vaccinee-related factors for vaccine success are a younger adult age [128], female gender [141], healthy state, a particular lack of primary or acquired immunodeficiency, favorable genetics [142], and gut microbiota [143].

However, accurate knowledge of the invariant antigen able to induce a long-term protective immune response and the precise identification of the immune correlates for protection, which are well known for a series of traditional vaccines [144] and have generally been identified in long-lasting bnAbs, is of extreme relevance. In COVID-19 vaccines too, post-mRNA vaccine-increasing titers of nAbs have been directly associated with increasing levels of vaccine efficacy, thus qualifying them as correlates of protection [145]. However, although nAbs seem to faithfully represent real correlates of protection, such as for COVID-19 vaccines and other infectious agents and traditional vaccines, and have been also used by the Regulatory Authorities for evaluating vaccine efficacy, key opinion leaders did not recognize nAbs as definitively acquired correlates of protection [146]. Recently, binding and nAbs have been confirmed as immune correlates of protection against the SARS-CoV-2 delta variant, indicating that IgG levels higher than 500 binding antibody units and nAb titers of 1024 or more may be considered thresholds for protection [147]. Nonetheless, many points still remain to be clarified, including the early waning of nAbs, and the role of specific cell-mediated immunity, which, at the experimental level has shown to be able of fully protecting from severe disease and death mice lacking nAbs [116], despite being unable to prevent breakthrough infections [148]. Both points should converge on the issue of the long-term maintenance of memory for bnAbs, which is crucial for the durability of vaccine-induced protection. It is now established that bnAbs are a minimal fraction of the repertoire established after infection or vaccination, and memory B-cells producing them can be lost without the persistence of GCs [94]. Moreover, new immunological pathways for modulating memory B-cells have been recently identified in chronic viral infections. In these conditions, memory B-cells seem to have no protective effect. However, a particular memory subset of B-cells enriched for interferon (IFN)-stimulated genes was recently found to be present in chronic viral infections, being responsible for an efficient, short-lasting antibody response. A strong IFN type I response early after infection may instruct this particular memory subset of B-cells [149].

## 8. Real-World Experience with mRNA Vaccines

The long period spent in the study of the biological characteristics of mRNAs made the development and emergency approval of mRNA vaccines by regulatory authorities possible in an unprecedented short time. The field test, which involved hundreds of millions of persons worldwide, was also of very short duration, making it record-breaking, never seen before in the face of an ongoing pandemic. However, questions remained whether the excellent safety and efficacy performance observed in the pre-registration studies would have been maintained in a much larger population, including immunosuppressed patients, such as onco-hematological, rheumatological, and transplanted patients, as well as pregnant women, who were lacking in the pre-registration studies. Moreover, the short observation time prevented the identification of possible long-term side effects. All these considerations, together with the appearance of immune-evasive viral variants, fueled a negative attitude against these innovative vaccines in large sections of the worldwide population. The Nobel Prize awarded to Katalin Karikò and Drew Weissman in 2023 for enabling the use of mRNA as a vaccine is the right recognition of the relevance of their discovery, which represents a real milestone that may be used for preventing and fighting infectious diseases and cancer.

### 8.1. Safety

Great concern was raised on the safety of mRNA vaccines, considering their innovative formulation and, therefore, totally unknown possible induction of short- and/or long-term side effects. Such a concern was fueled by several biological characteristics of the mRNA vaccines, which may have theoretically exposed vaccinated people to increased risk of autoimmunity, neurodegenerative diseases, infectious diseases, and cancer [150]. However, these concerns have been strongly criticized, particularly in relation to the presumed increased risk of cancer [151]. Moreover, the finding of mRNA molecules in different parts of the body, particularly the liver and spleen, for longer than expected, as long as 4 months after vaccine administration [152], adds further concern in relation to the possible induction of autoimmunity [153], even in the light of the high number of potential cross-reactivity between SARS-CoV-2 antibodies and human tissues [154]. However, real-world observation during the mass vaccination campaign did not confirm these concerns, considering that the severe side effects registered with slightly higher frequency than after traditional vaccines were anaphylaxis and myopericarditis. This latter adverse effect has not yet been definitely demonstrated to be due to an autoimmune reaction, being not associated with an increase in post-vaccine autoantibodies even in patients with autoimmune diseases who developed post-vaccine myocarditis [155]. Moreover, a Korean nationwide cohort study on 9,258,803 individuals concluded that no association between mRNA-based vaccination and increased risk for autoimmune connective tissue diseases could be established [156].

Although anaphylaxis has been rarely observed after mRNA vaccines, its frequency is slightly higher than 1.31 per million vaccine doses observed with traditional vaccines [157]. In one post-registration early study, a rate of anaphylactic reactions of 11.1 per million doses of Pfizer/BioNTech vaccine was observed, occurring within a median time interval from vaccination receipt to symptom onset of 13 min, mostly in females, with a median age 40 years, with prior documented history of allergies/allergic reactions and anaphylaxis. In the same study, even cases of less severe non-anaphylactic allergic reactions were reported [158]. The Centers for Disease Control and Prevention (CDC) monitored anaphylaxis after mRNA COVID-19 vaccines in the USA, in the period from 14 December 2020 to 18 January 2021, and found that out of a total of 9,943,247 administered doses of the Pfizer-BNT162b2 vaccine and 7,581,429 doses of the Moderna mRNA-1273 vaccine, 66 cases of anaphylaxis were reported, 47 of which followed the Pfizer-BioNTech vaccine (4.7 cases/million administered doses) and 19 following the Moderna vaccine (2.5 cases/million administered doses). No deaths occurred as a consequence of anaphylaxis. Also, in this study, cases were highly prevalent in young females with a prior history of allergy/anaphylaxis [159]. A prospective cohort study of the post-mRNA COVID-19 vaccine incidence of acute allergic reactions from 16 December 2020 to 12 February 2021 on 64,900 Mass General Brigham employees, who received their first dose of an mRNA COVID-19 vaccine, showed that self-reported acute allergic reactions were slightly, but significantly, more frequent with the Moderna mRNA-1273 vaccine compared with Pfizer-BNT162b2 (2.20% vs. 1.95%, *p* = 0.03), and anaphylaxis occurred at a rate of 2.47 cases per 10,000 vaccinations, generally in subjects with a history of previous allergic reaction or anaphylaxis, without significant differences between the two vaccines [160]. This rate of anaphylaxis was much higher than previously reported, but we need to point out that such self-reported solicited reactions were at variance with those in previous studies where the reporting was passive. Moreover, inaccuracies in the collection of information on the number of vaccine doses of the denominator cannot be ruled out [158,160], thus casting doubts on the absolute reliability of these data. However, it has been estimated that almost 4000 subjects with a history of severe food or drug allergy in that cohort have been safely vaccinated [158]. A recent study of anaphylaxis following the administration of COVID-19 mRNA vaccines allowed us to establish that the rate of anaphylaxis is higher than the rate observed after non-COVID-19 traditional vaccines, and it mainly occurs in young or middle-aged females with a history of previous allergic reactions/anaphylaxis, with the majority of cases occurring in the first 30 min after vaccine administration and more often after the first vaccine dose, with a generally favorable outcome and a safe administration of subsequent doses (Table 5). Another study reported similar rates of anaphylaxis after mRNA COVID-19 and traditional vaccines [161]. Despite hypersensitivity to PEG or polysorbate having been suggested as responsible for the reactions, the skin tests for PEG or polysorbate were of limited use to predict an anaphylactic reaction to the vaccine [162]. Many unexplained points still remain, including the pathogenetic mechanisms, since, when occurring at the first dose, the reactions may be non-IgE-mediated [163]. However, very recently, the pathogenetic mechanism was at least in part elucidated by identifying complement activation as the factor responsible not only for the inflammatory type 3 immunopathogenic reaction but also for basophil activation [164].

Myocarditis and/or pericarditis following vaccination with mRNA vaccines emerged early, in coincidence with the mass immunization campaign extended to the young and adolescents. In one study of over two million individuals (median age, 49 years) who received at least one dose of COVID-19 mRNA vaccines, acute myocarditis occurred with an incidence of 5.8 cases per one million individuals after the second dose [165]. A retrospective case series of US military vaccination in the period January–April 2021, when more than 2.8 million doses of the mRNA COVID-19 vaccine were administered, identified 23 men (median age, 25 years) who developed myocarditis, 20 of whom did so following the second dose, which is a rate higher than expected [166]. Data from the largest health care organization in Israel to evaluate the safety of the Pfizer-BNT162b2 vaccine confirmed an excess risk of myocarditis in the vaccinated. The control and vaccinated groups included 884,828 persons, and vaccination was associated with an elevated risk of myocarditis (risk ratio, 3.24; 95% CI, 1.55–12.44). SARS-CoV-2 infection was associated with an even higher risk of myocarditis (risk ratio, 18.28; 95% CI, 3.95–25.12) [167]. In another study of more than 2.5 million subjects above 16 years of age and vaccinated within the same health care system, 54 cases of myocarditis were recorded. This translates to an estimated incidence of myocarditis of 21.3 cases per 1 million individuals receiving at least one dose of the Pfizer-BNT162b2 vaccine, with the highest incidence among male patients of 16–29 years of age (106.9 cases per 1 million). Most cases were mild or moderate in severity [168]. Thus, the risk of myocarditis/pericarditis is increased following the administration of mRNA vaccines, particularly in adolescents and young people, mostly following the second vaccine dose, and generally with a mild clinical course and a favorable outcome [169] (Table 5).

Recently, a systematic review and meta-analysis on the relative risk of myocarditis and pericarditis in both those vaccinated against COVID-19 and those unvaccinated estimated that the relative risk for vaccination is increased two-fold, but the absolute risk is small, i.e., 0.8–16.7 cases per million in unvaccinated people versus 34.2 per million in vaccinated ones. Different pathogenetic mechanisms have been hypothesized, including mRNA-LNP-induced inflammation, the role of either of the sex hormones or human leucocyte antigens (HLAs), and the molecular mimicry between S protein and myocardial antigens; however, no hypothesis has been proven yet; thus, any pathogenetic mechanisms still remains elusive [170]. The post-vaccination risk of myocarditis/pericarditis had been previously described for smallpox vaccination and, to a much lesser extent, for the anthrax vaccine [171]; however, most vaccines are considered relatively safe, with a risk not significantly different from that of the unvaccinated population. On the basis of passive surveillance, smallpox vaccine-associated clinically diagnosed myopericarditis in over 540,000 smallpox-vaccinated US military personnel was estimated in approximately 124/million vaccinees [172]. However, a prospective study on military personnel revealed the new onset of chest pain, dyspnea, and/or palpitations during 30 days following either the smallpox or trivalent influenza vaccine in 10.6% and 2.6% of subjects, respectively [173]. The long-term effects of vaccine-induced myocarditis have not been assessed. Although precise epidemiological data are difficult to obtain and compare, it has been reported that COVID-19 infection-associated myocarditis is the most frequent severe complication, with an incidence of 1000–4000/100,000 COVID-19 patients per year and a survival of 30–80%, whereas “common viral myocarditis” has an incidence of 1–10/100,000 per year and a survival higher than 80%, and post-COVID-19 mRNA vaccination myocarditis has an incidence of 0.3–5/100,000 vaccinated people per year and a survival higher than 99% [174]. The risk seems to be affected by previous cardiovascular conditions or autoimmune diseases [175], and the frequency of cardiovascular complications at 18 months was lower for post-COVID-19 mRNA vaccination than for conventional myocarditis, whereas this was not the case for post-COVID-19 myocarditis [176]. The smallpox vaccine appears to pose the highest risk for post-vaccine myopericarditis induction [172], whereas, as far as the COVID-19 vaccines are concerned, the highest risk is observed in the mRNA ones, whose risk is however lower than that of the smallpox vaccine even though the most susceptible age group is considered (Table 5). Nonetheless, a recent retrospective longitudinal cohort multicenter study on 333 patients with COVID-19-vaccine-associated myocarditis from 38 US hospitals, while confirming an initial mild clinical course and a reassuring clinical outcome at a median follow-up of 178 days, allowed us to observe the persistence of late gadolinium enhancement in the cardiac magnetic resonance imaging in 60% of patients. This underlines the need for continued clinical surveillance and long-term studies in these patients [177]. In summary, post-mRNA vaccine myopericarditis may be generally observed after the second vaccine dose in young males, in whom it has been observed with an approximate frequency of 100/million vaccinees, a figure which is approximately 50–100 times higher than expected [178].

### 8.2. Effectiveness

The real-life effectiveness has been assessed in Israel by studying the response of newly vaccinated with Pfizer-BNT162b2 between 20 December 2020 and 1 February 2021. Each study group (vaccinated and unvaccinated controls) included 596,618 persons. The estimated vaccine effectiveness, at 7 days or more after the second dose, was 92% for documented infection, 94% for symptomatic COVID-19, 87% for hospitalization, and 92% for severe disease (ancestral Wuhan strain and Alpha variant). The estimated effectiveness in preventing death from COVID-19 14–20 days after the first vaccine dose was 72% [179]. A similar study from eight USA locations on health care workers found that ≥14 days after full vaccination, both Pfizer-BNT162b2 and Moderna mRNA-1273 were 90% effective in preventing symptomatic and asymptomatic COVID-19 infections [180]. The effectiveness for COVID-19-associated hospitalization at five Veterans Affairs Medical Centers involving 1175 subjects from 1 February to 6 August 2021 was 87%, similar to before (1 February to 30 June) and during (1 July to 6 August) the predominance of the SARS-CoV-2 Delta variant (84.1% and 89.3%, respectively). Protection was 80% for adults aged 65 years or older and 95% for those aged 18 to 64 years [181]. A case–control study of 3689 US adults at 21 US hospitals across 18 states during 11 March to 15 August 2021 reported 93% prevention of hospitalization for the Moderna mRNA-1273 vaccine and 88% for the Pfizer-BNT162b2 [182]. Another case–control study in Qatar during the period 1 January–5 September 2021 estimated that the Pfizer-BNT162b2 vaccine’s effectiveness against any SARS-CoV-2 infection was 77.5% in the first month after the second dose. Thereafter, the effectiveness gradually declined to approximately 20% 5 to 7 months later. Variant-specific (Beta and Delta variants) effectiveness waned with the same pattern. Conversely, the estimated effectiveness against any severe, critical, or fatal case of COVID-19 was 66.1% by the third week after the first dose and reached 96% or higher in the first 2 months after the second dose, persisting at approximately this level for 6 months [183]. A retrospective cohort study investigated the Pfizer-BNT162b2 vaccine’s effectiveness against SARS-CoV-2 infections and COVID-19-related hospital admissions in more than 3.4 million individuals between 14 December 2020 and 8 August 2021. For the fully vaccinated, effectiveness against SARS-CoV-2 infections was 73%, and against COVID-19-related hospital admissions, it was 90%. Vaccine effectiveness against infections of the Delta variant was 93% during the first month after full vaccination but declined to 53% after 4 months. The effectiveness against other (non-Delta) variants in the first month after full vaccination was 97% but waned to 67% at 4–5 months [184]. A retrospective cohort study on the Pfizer-BNT162b2 vaccine administered to over 83,000 patients after recovery from COVID-19 estimated a vaccine effectiveness of 82% for patients 16 to 64 years old and of 60% for patients 65 years of age or older [185]. A systematic review, while registering a great variability, could confirm an average marked (21 percentage points) reduction in protection for infection and a smaller reduction (average 10 percentage points) for severe infection in the period from 1 month to 6 months following full vaccination [186]. In a very large study, the Pfizer-BNT162b2 and Moderna mRNA-1273 vaccines were compared in terms of protection against infection, hospitalization, and death during a follow-up of 192 days in the time period of the dominance of the Alpha and Delta VOCs. The results showed that the absolute rates of infection, hospitalization, and death in both vaccine groups were low. However, “data suggests that compared to BNT162b2, vaccination with mRNA-1273 resulted in significantly lower rates of SARS-CoV-2-infection and SARS-CoV-2-related hospitalization” [187]. Conversely, no significant difference between the Pfizer BNT162b2 and Moderna mRNA-1273 vaccines in effectiveness for the prevention of COVID-19 hospitalizations on 1212 (593 cases and 619 controls) hospitalized patients in the period 11 March–5 May 2021 in the USA was observed. Overall, effectiveness was 87.1%, and full vaccination (two vaccine doses received at least 14 days before illness onset) was present in 8.2% of cases and 36.4% of controls, thus demonstrating the high vaccine effectiveness against hospitalizations for COVID-19 [188].

The quick spreading of the Omicron variant, carrying over 50 mutations, >30 in the S protein alone [183], and the rapid appearance of further subvariants, including BA.1, BA.2, BA.4/5, and XBB, raised the problem of their immune evasion ability due to poor recognition by vaccines and monoclonal antibodies [189]. This was clearly shown in real-world effectiveness studies, in which a three-dose Moderna-1273 vaccine protected against Delta infection 93.7% and 86% of vaccinees, at 14–60 days and >60 days, respectively, but only 71.6% and 47.4% against Omicron infection. The three-dose vaccine’s effectiveness against hospitalization was >99% with Delta or Omicron [190]. When the effectiveness of the Moderna-1273 vaccine was tested against Omicron subvariants, it was observed that the three-dose vaccine’s effectiveness against BA.1 infection was high and waned slowly, whereas the three-dose vaccine’s effectiveness against BA.2, BA.2.12.1, BA.4, and BA.5 infection was initially moderate to high and waned rapidly, thus seriously inducing a to consider the need for bivalent vaccines [191]. Taking advantage of the easy and quick adaptability of mRNA vaccines, bivalent vaccines for the ancestral Wuhan strain and the BA.4/5 subvariant have been developed and, when administered as boosters to persons who had previously been vaccinated, resulted in substantial additional protection against severe Omicron infection, with an effectiveness higher than that of monovalent boosters [192]. However, if the antibody response and effectiveness against the homologous strain were good, a lack of robust neutralization of other subvariants, BA.2.75.2, BQ.1.1, or XBB.1, was observed [193]. BQ.1.1 or XBB.1 were much more able to evade humoral immunity [194]. Nevertheless, this lack of antibody neutralization is compensated for by the maintenance of protective T-cell immunity [195] (Table 6).

However, the continuation of the Phase 3 COVE (Coronavirus Efficacy) study with open-label parts B and C up to April 2023 showed that “primary vaccination and boosting with mRNA-1273 demonstrated acceptable safety, effectiveness and immunogenicity against COVID-19, including emergent variants” [196], thus allowing the authors to conclude that even the monovalent vaccine with ancestral strain maintains a relevant protective role.

At any rate, a very promising pan-COVID-19 vaccine has been recently experimentally tested in hamsters, in a combination composed of an mRNA vaccine expressing the relatively conserved nucleocapsid protein of the ancestral SARS-CoV-2 virus with the S-based mRNA vaccine. Such a dual vaccine “confers complete protection against both BA.5 and BQ.1, preventing detection of virus in the hamster lungs. Analysis of respiratory immune response in mice shows that intramuscular mRNA-S+N immunization effectively induces respiratory S- and N-specific T-cell responses in the lungs and in bronchoalveolar lavage (BAL), as well as antigen-specific binding IgG in BAL” [197]. Should this vaccine maintain safety and efficacy even in clinical studies, it would represent a further step forward in the fight against COVID-19 with a universal vaccine, and a model for other diseases caused by highly variable RNA viruses.

The attempt to stimulate a waning immunity against SARS-CoV-2 with a fourth mRNA vaccine dose (second booster) has shown no safety issues associated with slightly increased protection against breakthrough infections, more markedly against symptomatic infections [198].

Very recently, mRNA vaccines have been shown to prevent the evolution toward long COVID-19 [199] as well as protect against several severe outcomes [200,201]. However, these findings may be subject to more detailed studies after the persistence of the virus is confirmed in the presence of dysfunctional T-cell immunity [202], even profiting from the opportunity that a universally accepted definition of long COVID has been recently agreed upon [203].

mRNA SARS-CoV-2 vaccine administration during pregnancy proved to be as safe and effective as in the general population [204], and the persistence in infants of vaccine-induced, trans-placentally transferred antibodies seems to be higher than the persistence of infection-stimulated antibodies [205]. Pregnant women vaccination, in addition to their own protection, was demonstrated to be an effective means for also protecting, through trans-placental antibody transfer, the infants not eligible for vaccination, particularly those aged <3 months [206].

### 8.3. mRNA Vaccines in Immunocompromised Patients

In pre-registration Phase 3 studies, immunocompromised patients were very few or lacking; thus, the different categories of immunocompromised patients were only investigated in real-world studies for safety, immunogenicity, and effectiveness, considering that in several countries, these categories were given access priority to vaccine administration by health authorities without clear knowledge of their actual risk of infection [207].

Among immunocompromised patients, those with immune-mediated inflammatory diseases (IMIDs) are at special risk of infections because they are not only exposed to the risk of infection like all the immunosuppressed patients but also to disease reactivation as a consequence of infection or vaccination, due to the hyperstimulation of the immune system [208]. The higher risk of severe and persistent COVID-19 infection in IMID patients has been confirmed; however, mRNA vaccines have been shown to be safe in these patients, but with reduced immunogenicity compared to healthy controls, owing more to the type of immunosuppressive therapy than to the type of IMID [202], with a behavior that appears not dissimilar from what has been observed with traditional vaccines in patients with IMIDs [209]. Rituximab and abatacept appeared as the most immunosuppressive treatments, but patients receiving mycophenolate, methotrexate, or Janus kinase inhibitors also showed reduced vaccination responses [208,210]. The safety of mRNA vaccines in IMIDs is expected, considering that mRNA is not infectious and patients with IMIDs are under immunosuppressive therapy, which may reduce both reactivity and immunogenicity. However, there is concern for possible symptom flares, due to the temporary interruption of immunosuppressive therapy in the occasion of vaccination.

In 449 SLE patients, Gerosa et al. found over 26% of local and/or systemic side effects after the first and/or the second shot. The most frequent side effects were fever, local reaction, fatigue, and arthralgia. Patients with constitutional symptoms and those under belimumab were more prone to develop side effects [211]. In 4% of the cases, a post-vaccine disease flare was observed. In another study on several hundred patients with different IMIDs, no vaccine-associated symptom exacerbation was reported [212], thus confirming that post-mRNA vaccine flares are rarely observed. Vaccine administration is; therefore, generally safe and well tolerated, provided that the guidelines released by the American College of Rheumatology [213] and/or EULAR recommendations [214] for the administration of these drugs in the safest conditions possible are carefully implemented [210]. In the guidelines, the timing of the interruption of some immunosuppressive drugs before and/or after vaccine administration is suggested, with the aim of reducing immunosuppression, thus facilitating an adequate immune response, but for a short time to prevent adverse events and flare ups. EULAR recommendations are less prescriptive and more conservative instead by delegating the management of the immunosuppressive therapy during COVID-19 infection or vaccination to the responsibility of the rheumatologist, who should always be involved [214]. The vaccine immunogenicity was adequate, but the seropositivity rate in IMIDs was significantly lower than in healthy controls; systemic glucocorticoids, rituximab, abatacept, and mycophenolate mofetil could impair immunogenicity [215]. The effectiveness of the Pfizer-BNT162b2 vaccine in autoimmune rheumatic diseases (ARDs) was good, particularly following the booster dose [216]. However, lower immunogenicity and effectiveness were observed in ARDs and healthy subjects with the appearance of the immune-evasive Omicron variant. Immunogenicity and effectiveness were more markedly reduced in ARDs than in controls [217]. Rituximab and incomplete vaccination were risk factors for breakthrough SARS-CoV-2 infection, and also incomplete vaccination for severe COVID-19 in patients with interstitial lung disease and systemic autoimmune disease [218]. In addition to the antibody response, even cellular immune response has been found to decrease [219]. In a Danish nationwide cohort study on over 150,000 patients with IMIDs under immunosuppressive therapy, followed up with in the period 1 January to 30 November 2021, immunosuppressive therapy was associated with an increased risk of infection and hospitalization, but not death. Anti-tumor necrosis factor (TNF), systemic corticosteroids, rituximab, and other immunosuppressants were particularly associated with these risks [220]. The immune response to vaccination in patients with IMIDs is not only lower than in normal controls but also less durable [221].

Although IMID and transplanted patients are theoretically at higher risk than patients with malignancy, considering that immunosuppressive therapy may hardly be interrupted in IMIDs and transplanted patients, malignancy chemotherapy is generally administered in cycles, for limited time periods, thus making it easier to plan the vaccine administration during chemotherapy-free periods, and we actually still do not know the difference in risk between different types of immunocompromised states [222]. An early analysis identified patients with malignancy and solid organ transplants as the categories at greater risk of severe COVID-19 and death [223]. COVID-19 continues to represent a real threat for immunocompromised people even in the Omicron era, as documented in a recent study in which immunocompromised patients accounted for 3.9% of the whole study population, but 22% of COVID-19 hospitalizations, 28% of COVID-19 ICU admissions, and 24% of COVID-19 deaths. The highest risk for COVID-19 hospitalization, ICU admission, and death was registered for solid organ transplant, followed by moderate to severe primary immunodeficiency, stem cell transplant, and recently treated hematologic malignancies [224].

A systematic review and meta-analysis established that only one third of solid organ transplant patients achieved seroconversion after full vaccination, with a progression to higher rates for hematological cancers, IMIDs, and solid cancers, and HIV patients with a seroconversion rate similar to that of immunocompetent controls. A third dose of an mRNA vaccine could stimulate seroconversion in non-responders with hematological cancers, IMIDs, and solid malignancy, whereas the response was variable in transplanted patients [225]. In 804 immunocompromised patients, including cohorts of immune-mediated rheumatic diseases, lymphoid malignancies, chronic renal disease, chronic liver disease, solid cancers, gastro-intestinal diseases under immunosuppressive therapy, and hematopoietic stem cell transplantation, specific antibody and T-cell immunity was checked after a third dose of either mRNA-1273 or BNT162b2 (administered to 751/804 patients). In the majority of these immunocompromised patients, a third vaccine dose improved the serological and T-cell response. Patients with chronic renal disease, lymphoid malignancy on B-cell-targeted therapies, or with no serological response after two vaccine doses were at higher risk of poor response to a third vaccine dose [226]. A comparative analysis of mRNA-1273 and BNT162b2 immunogenicity in immunocompromised subjects showed a significant higher prevalence of seroconversion and total antibody counts in mRNA-1273 than in BNT162b2, probably due to the higher dose of mRNA in the former [227]. In immunocompromised patients with systemic autoimmune diseases, hybrid immunity, coming from natural infection and vaccination, in analogy with what has been observed in normal people [109], makes the antibody response more robust and augments the cross-variant protection against COVID-19 [228]. A nationwide cohort study in Finland, Sweden, and Denmark on the comparative effectiveness of bivalent BA.4–5 or BA.1 mRNA booster vaccines among immunocompromised individuals allowed us to observe a lowered risk of COVID-19-related hospitalization and death over a follow-up period of 9 months [229].

Reeg et al. [230] studied T-cell immune responses to SARS-CoV-2 (either through infection or vaccination) in three groups of immunocompromised patients, specifically cancer (solid or hematological), HIV infection, and solid organ transplantation requiring robust immunosuppression. Significantly impaired SARS-CoV-2-specific immune responses after both natural infection and vaccination were observed. These patients displayed a functional state of cellular immunodeficiency that impairs T-cell responses; however, very few data points exist on a large scale. Their study provides evidence for the essential role of T-cells in enhanced protection against subsequent episodes of COVID-19. Therefore, studies analyzing only antibody responses may miss an essential component of the correlates of protection afforded by vaccination and overestimate the risk in immunocompromised patients. New evidence for this has been found in studies of primary humoral immunodeficiencies. A very recent report confirmed T-cell immunity in mice possessing B-cells and with normal lymphoid tissue architecture, yet devoid of both surface and circulating immunoglobulins. These mice resisted heterologous SARS-CoV-2 challenges post prior infection or mRNA vaccination. In the model, memory T-cells effectively reduced pathology but also early viral replication. Both CD4^+^ and CD8^+^ T-cells were required for this antibody-independent defense, with IFN-γ having a central role in the protection [116]. The authors noted that unlike nAbs, T-cell responses have long-term durability and also recognize a broad epitopic array in SARS-CoV-2 variants. A synergistic action between antibodies and T-cells is likely central for optimizing protective immunity and minimizing immunopathology.

However, the analysis of exposure to the conserved influenza antigen, M158–66 (GILGFVFTL), which generates a memory T-cell recall response with a functional memory repertoire that is polyclonal with the primary usage of the T-cell receptor (TCR) beta chain Variable19 gene in HLA-A2 individuals, showed that T-cell clonal expansion following vaccination in idiopathic juvenile arthritis patients receiving immunosuppressive therapy displays reduced clonal diversity compared to HLA-A2 healthy controls, with a phenotype that resembles that seen with aging [231]. These repertoire changes may have a functional repercussion, since the effector cytokine produced (IFN-γ) in such patients is reduced compared to the controls. Studies of T-cell immunity should complement serology to assess the real risk in immunocompromised hosts, since booster doses of vaccines may increase the breadth of T-cell responses with the reactivation of CD8^+^ cells and preservation of specific cellular memory [232].

Further hopes for an improved vaccine able to increase the response in vulnerable populations of immunocompromised subjects through an innovative adjuvant system come from a recent experimental study that used a lipid nanoparticle-encapsulated mRNA encoding IL-12p70 engineered with a multiorgan protection sequence to restrict transcript expression to the intramuscular injection site used (CTX-1796) with the Pfizer-BNT162b2 SARS-CoV-2 vaccine. This vaccine proved to be able to increase S protein-specific immune responses and extended their duration in mice [233].

## 9. mRNA Vaccines Against Other Infectious Diseases

### 9.1. Respiratory Syncytial Virus

Respiratory syncytial virus (RSV) is an RNA virus responsible for upper and lower respiratory infections in children ≤5 years old and elderly people, with an estimated global death toll of 100,000 in 2019 [234] and over 14,000, in 2015, respectively [235]. Fusion protein F is conserved; thus, it was chosen, in its prefusion conformation, as the target antigen in passive and active immunization, whereas protein G (attachment) is highly variable, allowing us to classify RSV in two subtypes, A and B, with the first one including 9 different genotypes and the other at least 32 genotypes [234].

Two monoclonal antibodies, Palivizumab and Nirsevimab, and two traditional recombinant vaccines have already been approved: GSK Biologicals Arexvy, containing 120 μg of pre-F recombinant protein and the adjuvant AS01_E_, indicated for ≥60-year-old people only; and Pfizer Abrysvo, composed of 60 μg of pre-F A and 60 μg of pre-F B antigens, without adjuvant, indicated for children, pregnant women, and elderly people. Both vaccines are administered as a single dose, are safe, and are efficient against RSV-related lower respiratory tract disease in ≥60-year-old adults by 82.6% [236] and 85.7% [237], respectively.

Moderna has developed an mRNA candidate vaccine against RSV, mRNA-1345, using the mRNA encoding the RSV prefusion glycoprotein F, which was very recently approved by the FDA and EMA for ≥60-year-old adults. The results of the Phase 1 study on the safety and immunogenicity of mRNA-1345 in 65–79-year-old volunteers were good, considering that the vaccine was well tolerated at all the tested doses (12.5, 25, 50, 100, 200 μg) and immunogenic, with antibodies persistently above the baseline throughout 12 months, when a booster dose was administered (NCT04528719) [238]. The Phase 2/3 study in ≥60-year-old adults (NCT05127434) showed that a single dose of 50 μg did not raise safety concerns and led to a lower incidence of RSV-associated lower respiratory tract disease and of RSV-associated acute respiratory disease than placebo among adults 60 years of age or older. Vaccine efficacy was 83.7% against RSV-associated lower respiratory tract disease with at least two signs or symptoms and 82.4% against the disease with at least three signs or symptoms [239]. This vaccine (mRESVIA) is the second mRNA vaccine approved against infectious diseases.

### 9.2. Influenza

Influenza is the classical viral infection that may benefit from the availability of an mRNA vaccine, considering the high variability of the virus, which implies the need to annually adapt the vaccine to the circulating viral strains. According to the World Health Organization (WHO), influenza is annually responsible for an estimated 1 billion infections at the global level, with 3–5 million severe cases and 290,000–650,000 deaths [240]. Four types of influenza virus are currently known, namely A, B, C, and D, where the first three are pathogenic for humans. Influenza A virus is highly mutant, with its frequent, even annual, drifts and more rarely periodical deeper mutations called shifts [19]. The frequent drifts impose the need to annually check for the best vaccine to be prepared based on the circulating viral strains, with the risk that a wrong prediction on the circulating strains may result in a vaccine failure for mismatch between the vaccine and circulating viral strains. Thus, quick and cost-effective vaccine production using the mRNA technique is extremely useful in the context of rapidly adapting the vaccine to the circulating strains, as a consequence of the frequent drifts or shifts. The relevant viral antigens to be included in the vaccine are the hemagglutinin (HA or H) and neuraminidase (NA or N). HA is used by the virus to enter cells with 18 different known subtypes in A viruses and 2 in B viruses, whereas NA lets the virus be released by the infected cells to infect other cells, amplifying infection in the host, with 11 different known subtypes in A and only one in B influenza virus [19,241]. HA subtypes are divided into two groups. Group 1 HA includes H1, H2, H5, H6, H8, H9, H11, H12, H13, H16, H17, and H18, whereas group 2 HA includes H3, H4, H7, H10, H14, and H15 [242]. Currently four different viral strains are circulating, namely A/H1N1, A/H3N2, B/Yamagata, and B/Victoria, even though B/Yamagata has not been isolated since March 2020 [243]. Thus, the seasonal influenza vaccines are either trivalent or quadrivalent. In addition to the high viral variability, which forces us to repeat vaccination annually, even the modest immune response to vaccine, with protection not higher than 60% in healthy adults [244], which wanes early [245], makes the prevention of influenza difficult to manage as well as limited and uncertain in the results.

Already in 1993, Martinon et al. immunized mice with an mRNA influenza vaccine in liposomes and observed the induction of virus-specific cytotoxic T-lymphocytes similar to those obtained with the infectious virus in mice of three distinct haplotypes, thus demonstrating cellular immunogenicity, which inactivated or subunit influenza vaccines scarcely obtain [246]. In 2022, three papers addressed the important issue of developing an mRNA universal vaccine against the highly mutant RNA influenza virus; two of them involved leveraging the use of a conserved invariant HA domain, such as the stalk domain [247]. McMahon et al., using an LNP-encapsulated mRNA quadrivalent vaccine containing HA stalk, NA, matrix protein 2, and nucleoprotein, from a group 2 influenza A virus, demonstrated that mice became broadly protected from morbidity after a single dose of 125 ng for each antigen [248]. The same group demonstrated the same protection in mice by administering a single dose of as low as 50 ng of each antigen of a pentavalent vaccine containing two B HAs from the two different strains, together with NA, matrix protein 2, and nucleoprotein [249]. Lastly, Arevalo et al. immunized mice and ferrets with a tentative universal influenza mRNA vaccine encoding all 18 type A and 2 type B HAs; they observed that the immunized animals were protected against a challenge with matched and mismatched influenza viral strains; such a protection was mediated, at least in part, by nAbs. These results demonstrate that the mRNA vaccine platform may effectively be used for inducing a protective immune response against multiple antigens in infections caused by variable viruses [250].

Moderna has already developed successful Phase 1 studies for monovalent H7N9 and H10N8 mRNA influenza candidate vaccines [251]. More recently, the company has even developed a seasonal influenza quadrivalent mRNA candidate vaccine, mRNA-1010, which is being tested in adults in Phase 3 (NCT05827978), after the completion of a Phase 1/2 study, of which its interim results appeared favorable as far as safety and immunogenicity are concerned. In the Phase 1/2 NCT04956575 study, mRNA-1010 was tested at 25, 50, 100, and 200 μg/dose; although reported adverse events were dose-dependent and generally more severe than those observed with an approved traditional quadrivalent vaccine, no safety concerns were observed. Immunogenicity was excellent at the doses of 25, 50, and 100 μg, with antibody titers higher than those observed with the traditional comparator vaccine as far as the response to viral strains of type A is concerned, whereas the response to type B strains was similar [252]. Even Pfizer started, in September 2022, a Phase 3 study on the safety, immunogenicity, and efficacy of an mRNA quadrivalent seasonal influenza candidate vaccine in subjects ≥18 years old (NCT05540522). In the Phase 2 study, a good CD4- and CD8-specific cellular immune response was documented. GSK/CureVac has instead completed a Phase 1 study on a quadrivalent seasonal influenza mRNA candidate vaccine (NCT05252338), and Sanofi Pasteur launched a Phase 1 study on a quadrivalent seasonal influenza mRNA candidate vaccine (NCT05553301). GSK/CureVac and Sanofi Pasteur have also activated Phase 1 studies for monovalent mRNA influenza candidate vaccines (NCT05446740 and NCT05426174, respectively).

### 9.3. Human Cytomegalovirus

Another mRNA candidate vaccine developed by Moderna is mRNA-1647 against human cytomegalovirus (CMV). Human CMV (also known as human herpesvirus 5) may infect the majority of normal individuals (approximately 60% of adults are infected in developed countries and up to 90% in developing countries) and establish a latency period, which is generally asymptomatic or pauci-symptomatic, in consequence of a strong host immune response. However, in immunocompromised patients, such as solid organ/stem cell transplant or human immunodeficiency virus (HIV)-infected patients, CMV may reactivate or reinfect, thus heavily conditioning the final outcome [253]. mRNA-1647 is composed of six mRNA molecules, five of which encode the subunits of the membrane-bound pentamer complex, and the sixth encodes the membrane-bound glycoprotein B. A Phase 1 study (NCT03382405) demonstrated that the vaccine was well tolerated and able to induce both humoral and cellular immune response [254]. Phase 2 (NCT04232280) has shown that, at a dose of 100 μg, the vaccine was safe and immunogenic in both CMV seropositive and seronegative participants [255], whereas a Phase 3 study on safety and immunogenicity in adult participants, and efficacy in seronegative female participants of childbearing age, is ongoing (NCT05085366 on the website clinicaltrials.gov, accessed on 20 August 2024). Moreover, a Phase 2, observer-blind, placebo-controlled, proof-of-concept trial for the evaluation of the efficacy, safety, and immunogenicity of the mRNA-1647 CMV vaccine in patients who have undergone allogeneic hematopoietic cell transplantation (HCT) is currently active and recruiting (NCT05683457 on the website clinicaltrials.gov, accessed on 20 August 2024). Preliminary results for the candidate human CMV mRNA vaccine look very promising, certainly better than those for the recombinant human CMV MF59-adjuvanted gB vaccine, which provided only approximately 50% protection. In particular, the antibody response as well as viral neutralization and antibody-dependent cellular cytotoxicity were stronger with mRNA than with the recombinant vaccine [256].

### 9.4. Zika

Zika virus was first described in Uganda in 1947; however, it became a problem for public health only when severe fetal malformations, such as microcephaly, were described during the 2015–16 epidemic in Brazil [257]. mRNA-1893, the candidate vaccine against the Zika virus, developed by Moderna, encodes the membrane and envelope protein (prM-E), inducing nAbs capable of preventing viral fusion. Protection against infection can be achieved, since only one serotype is known [17]. A Phase 1 study (NCT04064905) using 10, 30, 100, and 250 μg per dose has allowed us to observe the strong and persistent induction of nAbs both in seropositive and seronegative subjects and good tolerability with all tested doses, with the highest antibody geometric mean titers being observed with 10 and 100 μg [258]. Currently, a Phase 2 study (NCT04917861) has been launched and is currently ongoing, with the aim of evaluating the safety of two doses of the mRNA-1893 vaccine in over 800 flavivirus-seronegative and flavivirus-seropositive healthy adults compared with placebo [27,259].

### 9.5. HIV

For HIV infection, its end-stage disease is acquired immunodeficiency syndrome (AIDS), and an HIV pandemic was first described in 1981 among homosexual men in the USA [260], which, until an effective anti-viral therapy was developed in 1996, was invariably fatal and still continues to infect an estimated 1.3 million subjects worldwide each year and to kill too many people [261]. In 2021, 650,000 persons all over the world died as a consequence of HIV/AIDS [29]. So far, only three, and perhaps up to seven, HIV-infected subjects have been cleared from the virus, thanks to antiretroviral therapy coupled with CCR5 delta32/delta32 stem cell transplantation, not permissive to virus entry, but this therapy is very risky, and it cannot be imagined as a solution for the vast majority of cases [262]. Many unsuccessful efforts have been made in the past four decades for developing a safe and effective preventative vaccine. These have been unsuccessful because of the high virus variability and the closed trimer envelope conformation of the wild virus, characteristic of the most frequent phenotypes tiers 2 and 3 that are not neutralized by candidate vaccine-induced antibodies. These antibodies recognize the open and exposed trimer viral envelope, present in the tier 1A and, to a lesser extent, 1B, which are much less frequent, thus making these antibodies not broadly protective [133]. However, in a few subjects, broadly nAbs (bnAbs), capable of protecting against all HIV variants, have been found to emerge late in the course of the infection. These antibodies are produced by a small fraction of B-cells, which has been calculated to occur in the proportion of 1 for 2.4 million naïve B-cells [263], for an estimated total of 2700–31,000 naïve B-cells, approximately 15 for each lymph node, being precursors of the producers of bnAbs [264]. The recruitment of these precursor B-cells can be induced by a small recombinant molecule, eOD-GT8 60-mer, that mimics a small virus invariant protein needed for the virus to bind to and infect cells. It has recently been shown (NCT03547245) that two doses 8 weeks apart of 60 copies of eOD-GT8 60-mer on a nanoparticle scaffold of lumazine synthase in 36 volunteers are safe and induce a robust recruitment of precursor capable of synthesizing bnAbs in 97% of the immunized volunteers [265] and of helper T-cells in 84% of them [266]. This may be the first step of a multistep approach for a safe and effective HIV vaccine. Methods to enrich precursors of bnAbs secreting B-cells are being developed, and these also inform the development of new-generation vaccines [267]. Similar results were observed with an mRNA vaccine encoding eOD-GT8 60-mer (mRNA-1644, NCT05001373) [27,258]. Several methods have been reported to screen for bnAbs, which are a real hope for the prevention of the vertical transmission of HIV-1 as well as a promising avenue for treating SARS-CoV-2, influenza, and other viral diseases. They take a long time to develop, implying that some memory B-cells undergo further somatic hypermutation, and their late occurrence is probably due to gaps in affinity maturation [268].

### 9.6. Epstein–Barr Virus

Moderna started a Phase 1 study (NCT05164094) for an mRNA candidate vaccine (mRNA-1189) against Epstein–Barr virus (EBV) using four mRNAs encoding the envelope key entry complex glycoproteins gH, gL, gB, and gp42 [269] in seronegative adolescents 12–18 years old (part B) and in healthy adolescents and adults 12–30 years old (part A) [27]. EBV, also known as herpesvirus 4, is a γ-herpesvirus, and is the first human tumor virus identified. As with other herpesviruses, it has two phases of its life cycle, the latency phase, which may be life-long, and the lytic replication phase. It is estimated that 95% of the world’s population is infected; the infection occurs through saliva, and the virus may infect epithelial and B-cells. EBV may cause infectious mononucleosis; oral hairy leukoplakia in immunocompromised patients; autoimmune diseases, such as multiple sclerosis; cancers, such as naso-pharyngeal carcinoma and some types of gastric cancers; and lymphoproliferative disorders, such as Burkitt’s lymphoma, Hodgkin’s and non-Hodgkin’s lymphoma in transplanted and HIV patients, T-lymphoma, and natural killer (NK)/T-lymphoma [270]. It is, therefore, a virus that may induce a panoply of very severe pathologies, so an effective vaccine is needed.

### 9.7. Varicella Zoster Virus

Varicella Zoster Virus (VZV), also known as human herpesvirus 3, is a very contagious virus, transmitted through direct contact with the vesicles or by the inhalational route, which causes varicella, a vesicular exanthem that is generally a self-limiting, benign, disease, principally occurring before adolescence in Western countries in almost the whole population in the pre-vaccine era, and it generally lasts 5–7 days [271]. After varicella, VZV remains latent in the nerve ganglia, to be reactivated in >50-year-old individuals as herpes zoster or shingles, a disease characterized by a severe exanthema in a single dermatome and neuropathy, with severe pain. It may heal, but, particularly in aged immunocompromised subjects, it may also result in the highly disabling post-herpetic neuralgia, lasting even more than 1 year in 48% of the over 70-year-old population [272].

A live, attenuated vaccine for varicella has been licensed since the eighties of the last century in some European countries [271]; it is safe and effective. The same viral strain is used for a live, attenuated vaccine against herpes zoster, but it contains a higher amount of live attenuated antigen [271]. Although this vaccine was safe and effective too, it raised some concerns when considering that it would be needed for immunocompromised individuals, in whom living vaccines are contraindicated. Therefore, more recently, an adjuvanted recombinant traditional vaccine, the Shingrix, was developed. This is a subunit vaccine containing the glycoprotein E, which is important for viral replication, attachment, and entry [273]. This vaccine is highly effective, being able to prevent shingles in 100% of subjects, even those >70s years old [273].

Pfizer and BioNTech started, in 2023, a Phase 1/2 randomized, observer-blind study to evaluate the safety, tolerability, and immunogenicity of a modified RNA vaccine against VZV in healthy individuals of 50–69 years of age. This candidate vaccine is going to be administered in two doses, and its schedule will be tested in Phase 1 for use and then in a Phase 2 study (NCT05703607, clinicaltrials.gov accessed on 14 October 2024).

### 9.8. Other Pathogens

Moderna has already completed a Phase 1 study (NCT03392389) for an mRNA candidate vaccine (mRNA-1653) against the human metapneumovirus and parainfluenza 3, using two distinct mRNA sequences encoding the fusion proteins of the two viruses. This Phase 1 study, carried out on 124 healthy adults aged 18–49 years, who received two shots of different doses of mRNA-1653 4 weeks apart, has shown an acceptable safety profile and an increase in nAbs against human metapneumovirus and parainfluenza 3 [274]. Human metapneumovirus, of which A and B genotypes are known, belongs to the family of *Pneumoviridae*, and is responsible for upper and lower respiratory tract infections, and in 2018, it was credited as being responsible for an estimated 11,300 deaths worldwide among children ≤ 5 years old [275]. The human parainfluenza virus is another relevant cause of severe respiratory infections; four serotypes are known, and serotype 3 is the most frequent [274].

Moderna began a Phase 1 first-in-human study (NCT03325075) on an mRNA candidate vaccine (mRNA-1388) against Chikungunya RNA virus, which is transmitted by *Aedes aegypti* and *Aedes albopictus* mosquitoes and causes an illness characterized by fever, arthralgia, and skin rash. The candidate mRNA vaccine encodes the full Chikungunya virus’s structural polyprotein (capsid and envelope proteins E3, E2, 6 k/TF, and E1). Although the disease is self-limiting and the mortality is as low as 1%, arthralgia is incapacitating and may last up to one year in two thirds of patients. Chikungunya is present in over 100 countries, and neither therapy nor vaccine is available. mRNA-1388, administered to 54 volunteers, aged 18–49 years, in two scalable doses (25, 50, and 100 μg/dose) 4 weeks apart, was shown to be safe and immunogenic. It was well tolerated, and high-titer nAbs were observed to be long-lasting, up to one year post vaccination in the groups receiving the doses of 50 and 100 μg [276].

Lastly, Moderna has activated a Phase 1 study (NCT05398796) for an mRNA candidate vaccine (mRNA-1215) against Nipah virus, a virus first described in humans at the end of the nineties of the last century, which induces a zoonotic respiratory or neurologic disease with a very high fatality rate, so it has been classified a risk-4 pathogen [277].

Rabies is an almost invariably fatal disease caused by an RNA virus that is transmitted through the bite of both domestic and wild animals. The virus retrogradely proceeds along the nerves toward the spinal cord and the brain, where it induces fatal encephalomyelitis, which may manifest as furious or paralytic rabies. The WHO estimates 59,000 annual human deaths as a consequence of rabies, 95% of which are in Africa and Asia and approximately 99% following the bite of an infected dog. The prophylaxis of an at-risk wound may be carried out with passive and active immunization, which have long been available [278]. CureVac launched a Phase 1 study (NCT03713086) on a candidate mRNA vaccine (CV7202) against rabies using mRNA encoding the rabies glycoprotein; the study was successfully terminated and, after two doses of either 1 or 2 μg/each, it proved safe and immunogenic with both 1 and 2 μg/dose. All the volunteers showed antibody levels ≥0.5 IU/ml, which are considered adequately protective by the WHO [279].

Herpes simplex virus (HSV) types 1 and 2 are the causes of oral and genital herpes, respectively. They are present all over the world, with an estimated 67% of the world population infected with HSV 1 and 13% with herpes genitalis [280]. Like other herpesviruses, they live in latency, in the nerve cells, and may periodically be reactivated following different local or systemic stimuli [281]. BioNTech started a Phase 1 study (NCT05432583) for an mRNA candidate vaccine against HSV 1 and 2 (BNT163) in December 2022 on volunteer adults aged 18–55 years. It should be stressed that for all the herpesviruses, including human CMV, EBV, VZV, and HSV, the vaccines under study were tested for their efficacy not only as preventative vaccines but also in healthy adults already infected by the viruses in order to verify their possible efficacy as a therapeutic agent able to protect against a microorganism that has already entered the body.

BioNTech has also started Phase 1 clinical trials for mRNA candidate vaccines against tuberculosis in partnership with the Bill and Melinda Gates Foundation (NCT05547464) and malaria (NCT05581641), two big killers, considering that they were responsible for an estimated 1,600,000 [29] and 610,000 [282] deaths in 2021, respectively. A live vaccine against tuberculosis, named Bacillus Calmette-Guérin (BCG) after its discoverers, has been developed and used for over a century, even though its effectiveness is uncertain in adults [20]. In 2022, the Phase 1 clinical trial NCT05547464 was launched to evaluate the safety and immunogenicity of a tuberculosis mRNA-LNP candidate vaccine in BCG-vaccinated, HIV-negative adults, whereas the Phase 1 NCT05581641 clinical trial for malaria was launched to test the safety and immunogenicity of an mRNA-LNP vaccine encoding the circumsporozoite antigen of *Plasmodium falciparum* [258].

### 9.9. Combined mRNA Vaccines

Moderna launched a Phase 1/2 study on a combined influenza and SARS-CoV-2 mRNA candidate vaccine (NCT05375838) [283]. Moderna is even analyzing the candidate combined vaccines mRNA-1045 (influenza/RSV) in a Phase 3 study (NCT05330975) and mRNA-1230 (influenza/RSV/SARS-CoV2) in healthy 50–75-year-old adults in a Phase 1 study (NCT05585632). Pfizer/BioNTech activated a Phase 1 study on the safety and immunogenicity of a combined influenza and bivalent SARS-CoV-2 candidate vaccine (NCT05596734). Lastly, Sanofi Pasteur started a Phase 1 study on the safety and immunogenicity of an investigational combination mRNA vaccine for the prevention of RSV and/or human metapneumovirus in adults and older adults (NCT06237296).

Moreover, among 293 clinical trials on mRNA vaccines present on the website clinicaltrials.gov on 20 August 2024, five (NCT06028347, NCT04776317, NCT05876364, NCT06279871, and NCT06125691) refer to candidate self-amplifying vaccines against influenza and SARS-CoV-2, thus demonstrating that this promising way of producing effective vaccines with very low mRNA doses may become a reality, once safety and efficacy are conclusively demonstrated.

Table 7 summarizes the characteristics and the phase of study of approved and candidate mRNA vaccines, except the self-amplifying ones. The high number of reported studies demonstrate the interest in mRNA vaccines in the medical and scientific world.

## 10. Conclusions

mRNA vaccines represent a milestone in the history of vaccinology. The advantages of mRNA compared to traditional vaccines consist of their quicker, easier, and cheaper development, the consequent possibility of rapid adaptation to antigenic drifts and shifts, and in their ability to stimulate high levels of nAbs and specific cell-mediated immunity, which is seldom evoked with traditional vaccines. All these characteristics have been of paramount importance in the response to the COVID-19 pandemic with specific effective vaccines after just one year after its inception. In the pre-registration Phase 3 study, the efficacy against symptomatic infections was excellent, around 95%, as well as short-term safety, even though the population sample was relatively limited and immunocompromised subjects were not studied. The real-world experience of several hundreds of million people of different ethnic backgrounds and including immunocompromised patients substantially confirmed the excellent effectiveness results observed in the pre-registration studies. However, it quickly emerged that the protection was short-lived, no longer than a few months, and that the vaccine effectiveness decreased in relation to the early appearance of VOCs. While maintaining an acceptable protective level up to the Delta variant, a worrying reduction in protection occurred with the appearance of the Omicron variant and subvariants [284]. This is largely due to the virus with a mutation rate 10-fold higher than the influenza virus. A definitive solution of these issues for the majority of RNA viruses may only come from the setting up of universal vaccines [196,249,285,286], which are not available yet.

Regarding safety, although great concern was present when the Emergency Use Authorization was granted, considering that the mRNA vaccines had never been used before and many questions had no answers, particularly in relation to the possible increased risk of autoimmunity, what has actually been observed in the real-world studies is a slight increase in anaphylactic reactions compared to traditional vaccines, mainly in young and middle-aged women following the first vaccine’s administration, and of myopericarditis in young men, generally after the second dose’s administration, all of them with a favorable outcome, whereas more severe side effects were only anecdotal, thus without any statistical meaning. mRNA vaccines represent a step forward in vaccine preparation; however, the success of a vaccine depends mainly on the deep knowledge of the microorganism itself and its interaction with the immune system. In the case of COVID-19 mRNA vaccines, it is necessary to understand the reason(s) of the poor duration of the nAbs and to identify an invariant antigenic structure that is able to stimulate bnAbs in order to overcome the high variability of the SARS-CoV-2, a crucial issue shared with influenza. Moreover, a higher mRNA thermostability is desirable for safe use in countries where the cold chain is maintained with difficulty. Studies are ongoing to resolve this problem through different proposed techniques [36], and circular mRNA vaccines are promising in relation to this aspect. The safety and immunogenicity of Pfizer-BNT162b2 has even been found to be associated with specific gut microbiome markers, thus indicating a further possibility to improve tolerability and immunogenicity through targeted microbiota interventions [287]. The possible future development of a mucosal vaccine, capable of inducing a durable protective level of secretory IgA, may represent another tool to face the relevant issue of breakthrough infections [288], possibly through intranasal delivery [289].

## Figures and Tables

**Figure 1 vaccines-12-01418-f001:**
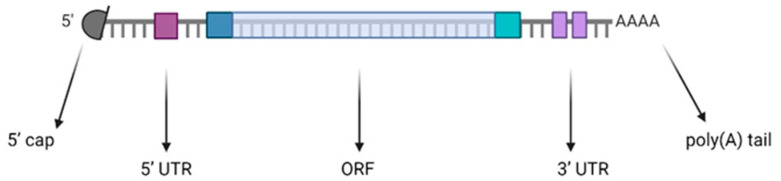
Schematic representation of synthetic mRNA. Created with BioRender.com.

**Figure 2 vaccines-12-01418-f002:**
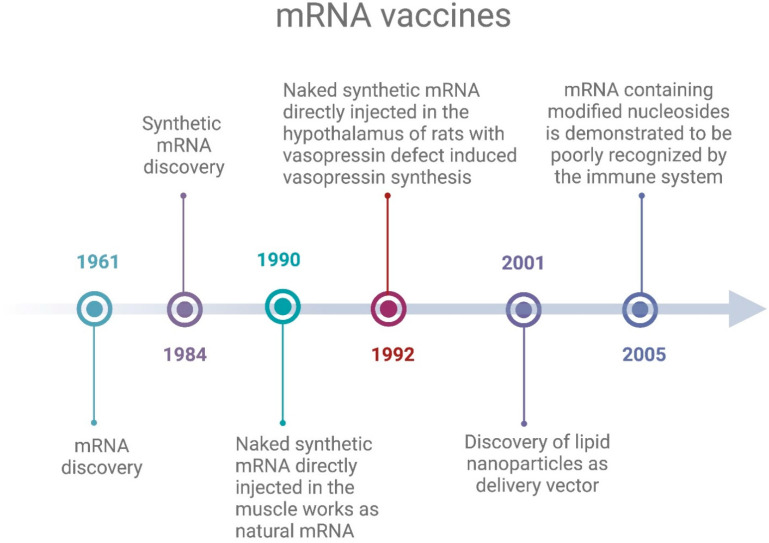
The main steps to mRNA vaccine development. Created with BioRender.com.

**Figure 3 vaccines-12-01418-f003:**
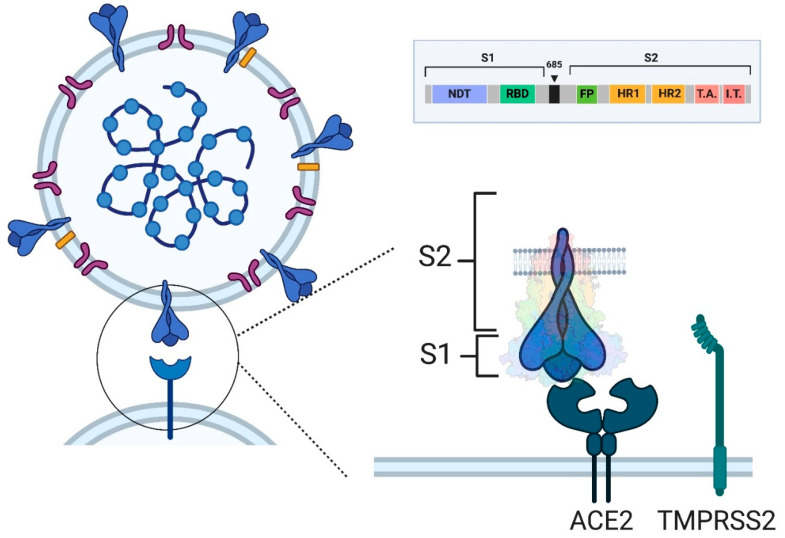
SARS-CoV-2 and protein S interaction with ACE2 and TMPRSS2 to enter cell. Created with BioRender.com.

**Figure 4 vaccines-12-01418-f004:**
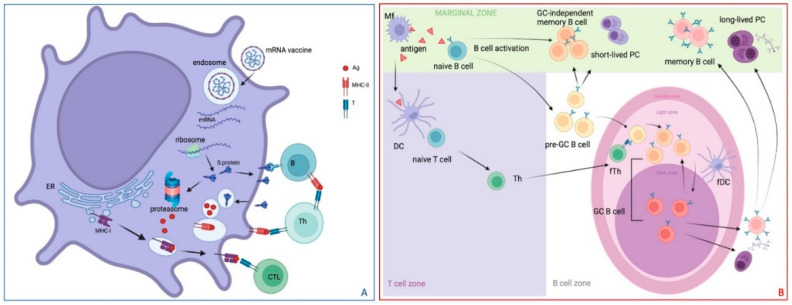
(**A**) The endocytic pathway of mRNA vaccine inside the antigen-presenting cell and its interaction with T-helper (Th) cells, cytotoxic T-lymphocytes (CTL), and B-lymphocytes; (**B**) the germinal center (GC) in secondary lymphoid organs and the extra-follicular pathways of B-cell maturation to memory B-cells and antibody-producing plasma cells (PCs). Created with BioRender.com.

**Table 1 vaccines-12-01418-t001:** Comparison between traditional (live attenuated, inactivated, and subunits) and mRNA vaccines (from ref. [10], modified).

	Types of Vaccines			
	Live, Attenuated	Inactivated	Subunits	mRNA
Development time *	Years	Years	Years	Months
Production cost	High	High	High	Low
Infection risk in vaccinees	Yes	No	No	No
Infection risk during manufacturing	Yes	Yes	Yes	No
Risk of integration	NA	NA	NA	Very low
Induced immunity	C/H	H	H	C/H
Safety	Caution in ID	Satisfactory	Satisfactory	Satisfactory
Efficacy	Generally high	Variable	Variable	H/S
Need for adjuvants	No	Yes	Yes	No

* Including the time for research, development, and commercialization. NA = not applicable; C/H = cellular/humoral; H = humoral; ID = immunodeficiency; H/S = high/short-term. No need for adjuvants because the delivery system (LNPs) and the mRNA itself act as adjuvants.

**Table 2 vaccines-12-01418-t002:** Genetic and functional characteristics of SARS-CoV-2 VOCs and current WHO designations (from refs. [34,37,45], modified).

WHO °	Pango ^@^	GISAID ^#^	Nextstrain ^^^	Documented ^†^	Designation ^‡^	S mut ª	Infect. ˜	Evasion ˢ	VOC ^˪^
Alpha	B.1.1.7	GRY *	20I/501Y.V1	Sep. 2020 UK ^˫^	18 Dec. 2020	8	29%	>2-fold	9 March 22
Beta	B.1.351	GH/501Y.V2	20H/501Y.V2	May 2020 SA ˹	18 Dec. 2020	8	25%	>8-fold	9 March 22
Gamma	P.1	GR/501Y.V3	20J/501Y.V3	Nov. 2020 BR ^⊕^	11 Jan. 2021	12	38%	NA	9 March 22
Delta	B.1.617.2	G/452R.V3	21A/S:478K	Oct 2020 IN ^⊗^	11 May 2021	9	97%	>5-fold	7 June 22
Omicron	B.1.1.529	GR/484A	21K	Nov. 2021	26 Nov. 2021	>30	NA	22-fold	14 March 23

° World Health Organization label; ^@^ Pango lineage; ^#^ Global Initiative on Sharing All Influenza Data clade; ^^^ Nextstrain clade; ^†^ date and place of first documentation: for Omicron, the place is represented by different countries; ^‡^ date of designation as a variant of concern; ª mutations in the protein Spike; ˜ increase in transmissibility; ˢ immune evasion as reduction in neutralization by sera of convalescent subjects; ^˪^ date of designation as previous variant of concern (Alpha, Beta, Gamma, Delta, Omicron); * formerly GR/501Y.V1; ^˫^ United Kingdom; ˹ South Africa; NA = not available; ^⊕^ Brazil; ^⊗^ India.

**Table 3 vaccines-12-01418-t003:** Characteristics of mRNA vaccines against COVID-19.

Project/Company	Encoded Antigen	Composition	Delivery System	Efficacy	Safety
Pfizer-BNT162b2	SARS-CoV2 Spike protein (S1 and S2)	mRNA 30 μg/dose	LNPs	95% protection	Good; slight increase in anaphylaxis myo/pericarditis in real-world studies
Moderna mRNA-1273	SARS-CoV2 Spike protein (S1 and S2)	mRNA 100 μg/dose	LNPs	94.1% protection	Good; slight increase in anaphylaxis myo/pericarditis in real-world studies

**Table 4 vaccines-12-01418-t004:** Factors able to influence the durability of vaccine-induced protection (from ref. [128], partially modified).

Vaccine-Related	Type of Exerted Influence
Platform	Live attenuated, VLP, and mRNA are the most effective platforms
Molecular conformation	Proteins are more effective than polysaccharide antigens—particulate and prefusion antigens are highly stimulating. Antibodies to the HIV trimer enveloped in closed conformation are broadly protective—adjuvants
Dose of antigen	The dose of antigen may modulate the recruitment of high-affinity antibodies and T-cell subpopulations
Schedule	ID and IM may differently modulate humoral and cellular immunity—higher intervals and heterologous prime-boost more effectively
Age	Elderly people are less responsive
Gender	Males are less responsive
State of immunity	Primary and secondary immunodeficiencies may be less responsive
Genetics	Human leucocyte antigens may influence T-cell responses significantly
Gut microbiota	It may influence the immunological response to vaccines

VLP = virus-like particle; HIV = human immunodeficiency virus; ID = intradermal; IM = intramuscular.

**Table 5 vaccines-12-01418-t005:** Summary of characteristics of adverse events in mRNA and traditional vaccines.

Adverse Event	Rate in mRNA Vaccines	Rate in Traditional Vaccines	Characteristics in mRNA Vaccines
Anaphylaxis	2.5–4.7/10^6^ doses	1.31/10^6^ doses	Young females, first dose
Myopericarditis	21.3 (106.9)/10^6^ persons *	124/10^6^ persons for smallpox vaccine	Young males, second dose

* In brackets is the rate of post-vaccine myopericarditis in the age group of 16–29-year-olds with the highest risk.

**Table 6 vaccines-12-01418-t006:** Effectiveness of mRNA COVID-19 vaccines in some real-world studies.

Viral Strains	Effectiveness Early After II Dose	Reference
Ancestral and Alpha	92–94% against infection; 87% against hospitalization; 92% against SD	[179]
Ancestral and Alpha	90% against symptomatic/asymptomatic breakthrough infections	[180]
Alpha, Beta, Delta	87% (80% ≥ 65 years—95% 18–64 years) against COVID-19 hospitalization	[181]
Alpha, Beta, Delta	88% (BNT162b2) and 93% (mRNA-1273) against COVID-19 hospitalization	[182]
Beta and Delta	77.5%, waning to 20% at 5–7 months; against SD 96% for 6 months	[183]
VOCs except Omicron	Delta: 93%, then 53% at 4th month; non-Delta: 97%, then 67% at 4–5 months	[184]
Omicron/subvariants	Specific humoral immunity: very reduced; specific cellular immunity: maintained	[191,195]

SD = severe disease; VOCs = variants of concern.

**Table 7 vaccines-12-01418-t007:** Characteristics of approved and candidate mRNA vaccines against other infectious diseases.

Project/ Company	Encoded Antigen	Composition	Delivery System	Phase	Efficacy/ Immunogenicity	Safety
mRNA-1345 Moderna	RSV F prefusion glycoprotein	mRNA 50 μg	LNPs	FDA/EMA-approved	82.4–83.7%	Good in Phase 1
mRNA-1851 Moderna	Influenza A H7N9	mRNA 25-50-75-100-400 μg	LNPs	Phase 1	Robust humoral immune response	Good
mRNA-1440 Moderna	Influenza A H10N8	mRNA 10-25-50 μg	LNPs	Phase 1	Robust humoral immune response	Good
PF-07252220 Pfizer	Quadrivalent A-B Flu strains	NA	LNPs	Phase 3	Documented specific CD4^+^ and CD8^+^ response	Good
CVSQIV GSK/CureVac	Quadrivalent A-B Flu strains	NA	NA	Phase 1	NA	NA
SP0273/MRT5407 Sanofi- Pasteur	Quadrivalent A-B Flu strains	NA	NA	Phase 1	NA	NA
mRNA-1647 Moderna	CMV 6 mRNA *	mRNA 50 μg	LNPs	Phase 3	Good, humoral, and cellular, in Phases 1–2	Good in Phases 1–2
mRNA-1644 Moderna	eOD-GT8 60-mer	NA	LNPs	Phase 1	NA	NA
mRNA-1893 Moderna	Zika prM-E	mRN 10-30-100-250 μg	LNPs	Phase 2	Strong, persistent nAbs with all the doses in Phase 1	Good in Phase 1
mRNA-1189 Moderna	EBV gH, gL, gB, and gp42	NA	LNPs	Phase 1	NA	NA
VZV modRNA Pfizer/BioNTech	VZV glycoprotein E	NA	NA	Phase 1/2	NA	NA
mRNA-1653 Moderna	hMPV/PIV3	NA	LNPs	Phase 1	NA	NA
mRNA-1388 Moderna	Chikungunya E3, E2, 6 k/TF, and E1	mRNA 25-50-100 μg	LNPs	Phase 1	High-titer nAbs lasting 1 year	Good
mRNA-1215 Moderna	Nipah	mRNA 25-50-100 μg	LNPs	Phase 1	NA	NA
CV7202 CureVac	RABV-G	1 or 2 μg/dose	LNPs	Phase 1	Good—all vaccinees had nAbs ≥ 0.5 IU/ml	Good
BNT163 BioNTech	HSV 1-2	NA	LNPs	Phase 1	NA	NA
BNT164 BioNTech-BMGF	Tuberculosis	NA	LNPs	Phase 1	NA	NA
BNT165b1 BioNTech	Malaria	mRNA PfCSP	LNPs	Phase 1	NA	NA
mRNA-1075 Moderna	Quadrivalent Influenza/ SARS-CoV-2	NA	LNPs	Phase 1/2	NA	NA
mRNA-1045 mRNA-1230 Moderna	Influenza/RSV Influenza/RSV/ SARS-CoV-2	NA NA	LNPs	Phase 1	NA	NA
PF-07252220 Pfizer- BioNTech	Quadrivalent Influenza/ SARS-CoV-2	NA	NA	Phase 1	NA	NA
Sanofi Pasteur	RSV/hMPV	NA	LNPs	Phase 1	NA	NA

* CMV = cytomegalovirus; 6 mRNA = encoding membrane-bound pentameric complex and glycoprotein B; LNPs = lipid nanoparticles; NA = not available; SARS-CoV-2 = severe acute respiratory syndrome coronavirus 2; FDA = Food and Drug Administration; EMA = European Medicines Agency; RSV = respiratory syncytial virus; prM-E = membrane and envelope protein; EBV = Epstein–Barr virus; gH = glycoprotein H, gL = glycoprotein L, gB = glycoprotein B, gp42 = glycoprotein 42; hMPV/PIV3 = human metapneumovirus and parainfluenza virus 3; E3, E2, 6 k/TF, and E1 = capsid and envelope proteins; nAbs = neutralizing antibodies; RABV-G = glycoprotein of the rabies virus; HSV = herpes simplex virus; BMGF = Bill and Melinda Gates Foundation; PfCSP = *Plasmodium falciparum* circumsporozoite protein.

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
