# Peer review of "mRNA Vaccines Against COVID-19 as Trailblazers for Other Human Infectious Diseases"

_vaccines, 2024, doi:10.3390/vaccines12121418_

Round 1

Reviewer 1 Report

Comments and Suggestions for Authors

The paper comprehensively addresses various aspects related to the introduction of an mRNA vaccine into widespread use in connection with the SARS-Cov-2 virus pandemic. This particular virus receives almost all of the attention, and the vast majority of the information in the manuscript refers to this virus and the mRNA vaccines developed against it. Hence, the reviewer suggests that the title of the paper should more closely reflect this particular area of information. The section devoted to experimental mRNA vaccines against other microorganisms could, of course, remain, but as a supplement rather than an equal component. Reading the paper gives the impression that the authors are characterized by a rather enthusiastic approach to this new category of vaccines. This, of course, is not a reproach, after all, in the paper one can easily find references to the objections and doubts that arise, although the manner in which they are dispelled is not entirely convincing. Page 4 (lines 135-136) mentions that mRNA is characterized by a lower risk of integration into host DNA than is the case with DNA vaccines. And this is one of the primary concerns in the public perception of this category of vaccines. It would therefore be appropriate to include specific information on what this risk is, whether such a process has been experimentally demonstrated, and if so, what the consequences were. More attention should also be paid to the long-term effects of immunization. No reference was made in any way to the fact that despite more than 30 years of mRNA vaccine research, no mRNA vaccine had been approved for marketing by the time of the pandemic. The anti COVID-19 vaccine was authorized on an emergency procedure, which was mentioned only once, in the conclusions (line 1336) without providing any justification for this decision. From the context of this part of the paper, one might understand that the reason for this decision was concerns about autoimmune reactions and cardiac complications (myopericarditis), yet one of the manufacturers of the mRNA vaccine has only relatively recently disclosed the possibility of the latter consequences. Cancer complications were also not addressed in sufficient detail. A recent article, “Increased Age-Adjusted Cancer Mortality After the Third mRNA-Lipid Nanoparticle Vaccine Dose During the COVID-19 Pandemic in Japan” (DOI: 10.7759/cureus.57860), which, although retracted, provides some very interesting information, and the reason for the retraction itself is not entirely clear. Moreover, the completely “reassuring” tone regarding complications of myocardial injury also seems unjustified because the authors of a recent publication in EClinicalMedicine (Part of the Lancet; 2024 Sep 6:76:102809. doi: 10.1016/j.eclinm.2024.102809) state that “...despite a mild initial course and favorable mid-term clinical outcome, warrants continued clinical surveillance and long-term studies....” 
The above remarks do not discredit the evaluated work, which the reviewer considers valuable, they are only to draw attention to certain aspects that are not zero-one, and which should be treated a little more carefully.

Reviewer 2 Report

Comments and Suggestions for Authors

This manuscript is a extensive review of mRNA vaccines. Is a topic of great interests, therefore I consider that this publication can be of help for healthcare professionals, University professors, etc.

I think that the manuscript could be improved in the way of organization of the different sections. First in the section devoted to mRNA, I think that the delivery systems should be included in a subsection or separate section.

The section devoted to SARS-CoV-2 could go before the section of mRNA vaccines that include potential vaccines for other viral infections. As COVID-19 vaccines are the main topic of the manuscript I think that all the information about it should go first and after the other possible uses of mRNA vaccines.

In relation to delivery systems:

-          What cationic lipids are used?

-          What cholesterol components are used?

-          Cholesterol contribute to the stability of LNP, but also cholesterol contributes to regulate the rigidity of the membrane.

Reviewer 3 Report

Comments and Suggestions for Authors

This epic review  is of considerable importance in explaining the development of mRNA vaccines and providing updates on the application to infection beyond COVID. It is however a review in two parts, first mRNA vaccines in general and second mRNA vaccines application to COVID. It might be better to be divided into two parts so the covid section can be adequately introduced and summarised and indexed.

A general point that is missing is knowledge of the cells that produce the vaccines encoded antigens and fate of the antigen including systemic release.  Is this a subject that needs more investigation?

With respect to safety (and this is a subject the first sentence of the abstract) it is recommended that consideration be given separately to the mRNA technology in general and then to its use for specific proteins, especially relevant for the disease-causing covid spike protein. The profile of adverse effects in the non-covid vaccine trials would be very important here (i.e. are the adverse effects found for covid all due to the spike protein?)

An important aspect of the mRNA vaccines is the possible deleterious effects of the multiple short-interval  boosters that were used. The reduction of the generation of diversity of the antibody responses and original antigen sin has been reported and at least for influenza (subunit vaccine) increasing ineffectiveness after multiple boosts.

The paper is silent on the covid vaccination of children. 
